# Learning produces an orthogonalized state machine in the hippocampus

Weinan Sun[1,2,5 ✉], Johan Winnubst[1,5], Maanasa Natrajan[1,3,4], Chongxi Lai[1], Koichiro Kajikawa[1], Arco Bast[1], Michalis Michaelos[1], Rachel Gattoni[1], Carsen Stringer[1], Daniel Flickinger[1], James E. Fitzgerald[1,4] & Nelson Spruston[1 ✉]

Cognitive maps confer animals with flexible intelligence by representing spatial, temporal and abstract relationships that can be used to shape thought, planning and behaviour. Cognitive maps have been observed in the hippocampus[1], but their algorithmic form and learning mechanisms remain obscure. Here we used large-scale, longitudinal two-photon calcium imaging to record activity from thousands of neurons in the CA1 region of the hippocampus while mice learned to efficiently collect rewards from two subtly different linear tracks in virtual reality. Throughout learning, both animal behaviour and hippocampal neural activity progressed through multiple stages, gradually revealing improved task representation that mirrored improved behavioural efficiency. The learning process involved progressive decorrelations in initially similar hippocampal neural activity within and across tracks, ultimately resulting in orthogonalized representations resembling a state machine capturing the inherent structure of the task. This decorrelation process was driven by individual neurons acquiring task-state-specific responses (that is, 'state cells'). Although various standard artificial neural networks did not naturally capture these dynamics, the clone-structured causal graph, a hidden Markov model variant, uniquely reproduced both the final orthogonalized states and the learning trajectory seen in animals. The observed cellular and population dynamics constrain the mechanisms underlying cognitive map formation in the hippocampus, pointing to hidden state inference as a fundamental computational principle, with implications for both biological and artificial intelligence.

Intelligence, at its core, manifests in the ability of an organism or agent to engage dynamically with its environment, interpret information, adjust to unfamiliar situations and execute complex tasks. A central concept in the study of natural and artificial intelligence is the notion of an 'internal model'. These models convert external world observations into a well-organized representation, thereby enabling adaptive behaviour. In neuroscience, a notable example of an internal model is the concept of a 'cognitive map'. Conceptualized early in the twentieth century, cognitive maps are neural constructs that enable animals to comprehend their environment and understand the interactions between their bodies and the external world, which supports efficient navigation, even in novel circumstances[2]. This concept gained momentum with the discovery of 'place cells' in the hippocampus, neurons that fire selectively at specific positions in an environment[3,4]. Since then, the neural underpinnings of cognitive maps have been studied extensively, revealing a vast body of knowledge about the firing properties of neurons that comprise cognitive maps in the brains of rodents[5], primates (including humans)[6] and other animals[7,8].

These foundational studies indicate that the hippocampus not only captures features of the environment but also the relationships between them and the actions of the animal within it. For example, many hippocampal neurons carry information in the form of activity that is greatest at a particular location in the environment (the 'place field' of the cell)[3], whereas others store information not only about place but also about contextual features such as the movement direction[9,10], running speed[9] or movement history[11–13] of the animal. Hippocampal neurons can also learn to represent more abstract spaces, such as the position in a sound landscape[14], accumulated evidence[15], arbitrary relationships between concepts, objects or events[16], and other non-spatial dimensions[17,18]. Despite an extensive body of knowledge about the neuronal firing properties constituting hippocampal cognitive maps, and recent ideas concerning their algorithmic structure[19–22], we are still yet to fully characterize the formation of cognitive maps during the entire learning phase of moderately complex tasks. Acquiring such empirical data is crucial, and technical advances, such as increasing the number of recorded neurons and the duration of longitudinal tracking, can facilitate our ability to study the formation of cognitive maps representing

[1]Janelia Research Campus, Howard Hughes Medical Institute, Ashburn, VA, USA. [2]Department of Neurobiology and Behavior, Cornell University, Ithaca, NY, USA. [3]Department of Neuroscience, Johns Hopkins University, Baltimore, MD, USA. [4]Department of Neurobiology, Northwestern University, Evanston, IL, USA. [5]These authors contributed equally: Weinan Sun, Johan Winnubst. ✉e-mail: sunw37@gmail.com; sprustonn@hhmi.org

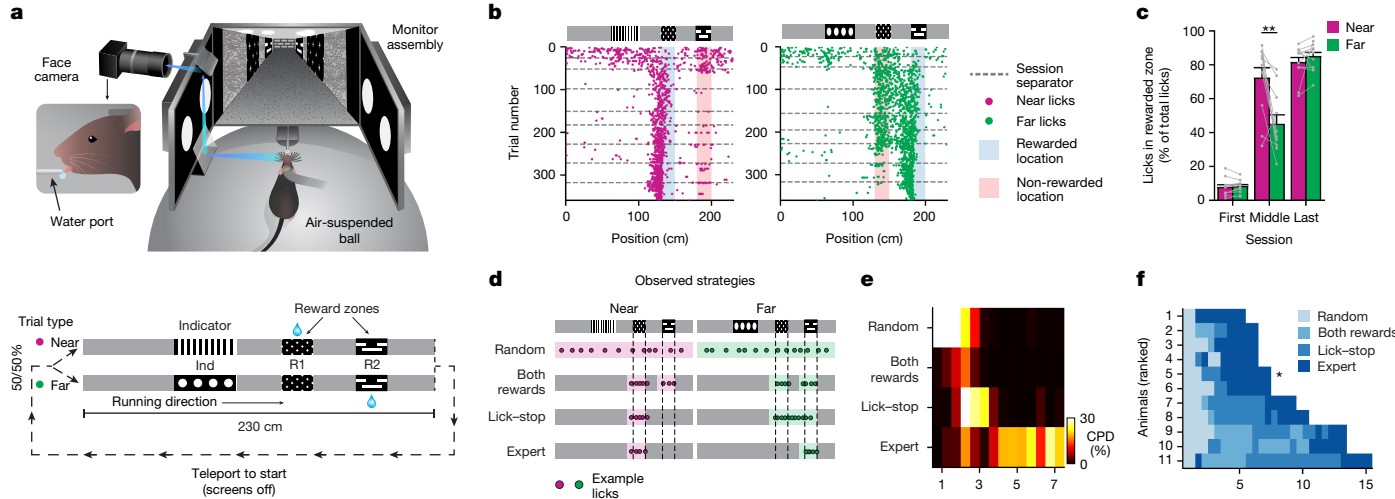

**Fig. 1 | Mice exhibit systematic behavioural strategy changes when learning the 2ACDC task. a**, Diagram of the virtual reality behavioural setup (top) and illustration of the task (bottom). The diagram of the virtual reality behavioural setup was created by J. Kuhl. **b**, Example lick patterns of a single mouse across sessions for both the near (magenta) and far (green) trial types. Each dot represents a single lick within a trial, and the dashed lines indicate separation between daily sessions. **c**, Percentage of licks at the correct reward locations for both the near and the far trial types for the first session, an intermediate session and the last session in which the mice show expert performance ($n = 11$ mice; $**P = 0.004$, two-sided, paired Student's $t$-test;

bar graph showing mean ± s.e.m.). **d**, Illustration of four different behavioural strategies that mice exhibit during learning. The coloured shadings denote the non-zero regions within the basis functions for regression against licking density across track locations. **e**, Coefficients of partial determination (CPDs) for the four behavioural strategies for one example mouse. **f**, Dominant strategy for all mice over sessions (ranked by learning speed). The asterisk denotes the mouse shown in **e**. The values in **e,f** are calculated by splitting each session into two parts of equal duration (start and end). For the number of sessions per strategy (mean ± s.e.m.), 1.9 ± 0.7 (random), 2.0 ± 2.5 (both rewards), 3.3 ± 2.8 (lick–stop) and 3.6 ± 0.5 (expert).

complex relationships that require extensive exploration and learning. Here we leveraged such technological advances to follow neural activity in thousands of hippocampal neurons in each mouse, stably for many days or weeks, as they learned to perform a task requiring them to form cognitive maps representing spatial, temporal and abstract relationships while interacting with a complex but predictable environment.

Our results show that during learning, mice proceed through a stereotypical series of behavioural changes that are mirrored by structured changes in neural activity. Specifically, hippocampal activity undergoes a series of decorrelation steps that orthogonalize neural activity in regions of the environment where sensory stimuli are similar but underlying task states differ. We analysed the representational structure of the task, visualized the low-dimensional geometry of the neural activity and compared neural activity in the hippocampus to unit activity in various cognitive models and artificial neural networks trained on the task. We show that day-to-day dynamics of hippocampal activity are consistent with the formation of a state machine, consisting of orthogonalized latent representations of task states. The transitions between these latent states, each encoding specific task features or segments, are learned to predict the dynamics of the interaction of the animal with the environment. Orthogonal states can represent similar sensory stimuli over learning, highlighting latent task structure as the driver of state orthogonalization.

We show that the final orthogonalized state machine (OSM) representation observed in the hippocampus can be reproduced by several computational models, including a type of hidden Markov model (HMM) called clone-structured causal graph (CSCG)[21,23] and certain recurrent neural networks (RNNs). Although various models can achieve the final orthogonalized representations through specific architectural designs or learning objectives, CSCG uniquely replicates both the end-state and the step-by-step learning trajectory observed in animals. This alignment between the learning dynamics of the CSCG and hippocampal activity patterns suggests that latent state inference processes, as implemented in the CSCG, may be fundamental to understanding the principles of hippocampal learning and cognitive map formation.

By contrast, despite their effectiveness in sequence modelling, popular models such as long short-term memory (LSTM)[24] or transformers[25] do not naturally generate representations that mirror those observed during animal learning. We further demonstrate that neural activity shows flexible adaptations of the OSM in altered task conditions such as the introduction of new visual cues and adjustment in the lengths of track segments. In summary, these findings shed light on the computational principles governing the formation of cognitive maps and provide potential guides for the design of future artificial systems.

## Learning the two-alternative cue–delay–choice task

We trained transgenic mice expressing GCaMP6f in hippocampal neurons to navigate in a virtual reality environment while head fixed to enable imaging of neural activity as they learned the relationship between visual cues and the future location of water reward delivery in two linear tracks (Fig. 1a; see Methods for details). On each trial, water was delivered at one of two reward zones, either near or far from the beginning of the track, which we called R1 and R2, respectively. Before these rewarded locations, a visually distinct indicator cue (Ind) perfectly predicted the rewarded location (Fig. 1a, bottom, and Supplementary Video 1). Efficient execution of this two-alternative cue–delay–choice (2ACDC) task requires mice to form and use long-term memory of the relationship between the indicator cues and reward locations, and short-term memory of the indicator cue after it disappears and before the rewarded location.

Mice were initially trained for 5 days (a 1-h session each day) to run on a spherical treadmill and collect randomly delivered water rewards in the dark. Subsequently, screens were turned on to display the virtual reality environment. In each subsequent 1-h daily session, mice performed approximately 80–200 trials (124 ± 43 trials, $n = 11$ mice), with the reward location depending on the trial type. Both near and far trial types were presented sequentially in a randomized manner (Methods). To initiate a new trial, mice had to run to the end of the corridor, which was decorated by a 'brick wall' cue, and a 2-s period

of dark screens ('teleportation') preceded the next trial. The two trial types shared identical visual cues at all locations except the indicator region. Outside of the indicator region and the reward regions, the walls of the virtual corridor were decorated with relatively featureless grey wood grain ('grey' regions). This sensory ambiguity within trials (four grey regions) and across trials (both grey regions and reward-zone cues are visually identical) is a key feature of the task. For the first 1–3 days of training, water rewards were delivered even if the mouse did not lick in the rewarded zone, until consistent anticipatory licking was observed. On all subsequent days, mice were rewarded with a drop of water on any trial only if they licked in the correct reward zone. No penalty was imposed for licking in other locations. Thus, mice learned the task through exploration, presumably motivated to slow down and lick only when rewards were expected.

We assessed learning by plotting licking behaviour as a function of position for all trials across several days of training on the 2ACDC task (Fig. 1b). Initially, mice licked throughout the entire track, but they quickly learned to restrict licking to portions of the track near the two reward zones in both trial types. This change in behaviour occurred within 2–3 sessions in all mice (Extended Data Fig. 1a). Around the same time, mice developed an intermediate strategy and learned to suppress licking after receiving a reward. As a result, licking behaviour was close to optimal for the near trial type (not licking at the far reward zone), but remained suboptimal for the far trial type (Fig. 1c, middle session), because licking began at the near reward zone and was often sustained until the reward delivery at the far reward zone. With additional training, mice eventually learned to suppress licking in the near reward zone on the far trial type, thus achieving close to optimal performance on both trial types (Fig. 1c, last session). Thus, licking behaviour appears to evolve through gradually shifting phases, characterized by dominant behavioural strategies that change over time (Fig. 1d): (1) random licking, (2) licking in both reward locations, (3) licking in reward locations and stop licking after a reward is collected ('lick–stop'), and (4) only licking near the correct reward locations ('expert'). These strategies represent predominant behaviours that emerge and fade gradually, rather than discrete, abrupt changes.

We used a statistical method called coefficient of partial determination to assess the contribution of each of the four behavioural strategies to the overall behaviour of the mice. Using these four behavioural strategies as regressors accounted for $36.5 \pm 5.9\%$ of variance in licking behaviour averaged across all sessions ($n = 9 \pm 3$ sessions per mouse). This explained variance percentage is within the expected range for behavioural studies involving complex tasks[26]. By removing a regressor corresponding to each behavioural strategy one at a time, we were able to determine its unique contribution to the behaviour. Coefficient of partial determination analysis demonstrated that these four strategies emerged and became dominant in successive waves at different points during the learning process (Fig. 1e), which was mirrored by gradual changes in the profile of the running speed of mice (Extended Data Fig. 1b). Despite variations in the number of sessions required for different mice to reach expert performance, the gradual progression through these dominant behavioural strategies was consistently observed (Fig. 1f).

## Imaging of hippocampal activity during learning

Before training, all mice were implanted with a cranial window to enable imaging of neural activity using GCaMP6f expressed in pyramidal neurons in area CA1 of the dorsal hippocampus (Fig. 2a and Methods). Activity was imaged using a two-photon random access mesoscope[27]. The 3-mm cranial window was readily imaged with the two-photon random access mesoscope, which has a 5-mm field of view. Thousands of cells (within single sessions: $4,682 \pm 827$ on average per mouse across all sessions, range: 3,813–6,490; maximum single session cell count: $5,545 \pm 848$ on average per mouse, range: 4,266–7,309; $n = 11$ mice)

mostly near the centre of the field of view, were readily resolved and reidentified in each session across several weeks of training and imaging (cells tracked across sessions: $3,954 \pm 661$ on average per animal, range: 3,034–5,354; Methods, Fig. 2b and Extended Data Figs. 2 and 3).

By the second session, many cells had increased activity at well-resolved positions along the track. Ordering cells according to the position of the peak activity revealed a clearly resolvable diagonal band of spatial responses tiling the whole virtual track in both trial types (Fig. 2c). Ordering cells according to their spatial activity pattern for the opposite trial type also showed a diagonal spatial band, indicating that many cells are active at similar locations in both trial types. However, activity differed between trial types most prominently at the indicator cue position (Fig. 2c), indicating that sensory information dominates the neural activity at this early learning stage. Indeed, cells that were most active in one of the four grey regions also showed moderate activity in other grey regions (Fig. 2c). We observed considerable individual variability in the initial representation of the task across mice, with some showing strong decorrelation from the very beginning, whereas others exhibited high correlation (Extended Data Fig. 4). After several more days of training, these neural activity-based 'maps' of the 2ACDC task increasingly differentiated within single trial types (among several grey regions) and between corresponding locations across the two trial types (Fig. 2d).

## Systematic hippocampal changes during learning

The representational structure of the near and far trial types was compared by computing a population vector correlation for the two trial types (Methods), which decreased systematically in selected positions during training. Analysis of the cross-correlation between near and far trial types across all regions of the track indicated that the indicator cue region had low correlation, as expected from the differences in visual stimuli at this location (Fig. 2e,f). Both within and between trial types, the four grey regions of the track were moderately correlated on the first session for most mice, but by the third session, the correlation was significantly reduced (Fig. 2e,f and Extended Data Fig. 5 showing population vector angles also approaching 90°), suggesting that the hippocampus had orthogonalized its representations of these visually similar regions. Correlations between corresponding locations across trial types decreased in an ordered manner, with the neural activity at the track region right before the far reward zone (pre-R2) decorrelating generally earlier than the region before the near reward zone (pre-R1; Fig. 2e,f). Neural activity corresponding to the indicator cue, while already showing low correlation from the onset, decorrelated further with increased exposure to the two trial types (Fig. 2e,f). Although most track regions underwent complete decorrelation into near orthogonal representations, the beginning and end of the track remained correlated throughout training for most mice (Extended Data Fig. 4). This suggests that the decorrelation process is shaped by the task structure, as after collecting the reward, the animal lacks information about the next trial type until seeing the next indicator.

Together, these results reveal a systematic progression in how the hippocampus learns to represent the task structure. Initially, the hippocampus differentiates each of the four visually similar grey regions within individual linear tracks, suggesting that the hippocampus first learns the sequential nature of the task environments. With additional training, neural activity at corresponding locations across the two trial types progressively decorrelates, generally beginning with the region before the far reward, followed by the region before the near reward. This progressive decorrelation towards orthogonalization occurs despite identical visual cues, suggesting the emergence of distinct task state representations. This gradual decorrelation in neural activity co-evolves with the progressively improving licking behaviour of the mouse (Extended Data Fig. 6). This average learning trajectory (Fig. 2f) was observed in most individual mice (for example, Fig. 2e); however,

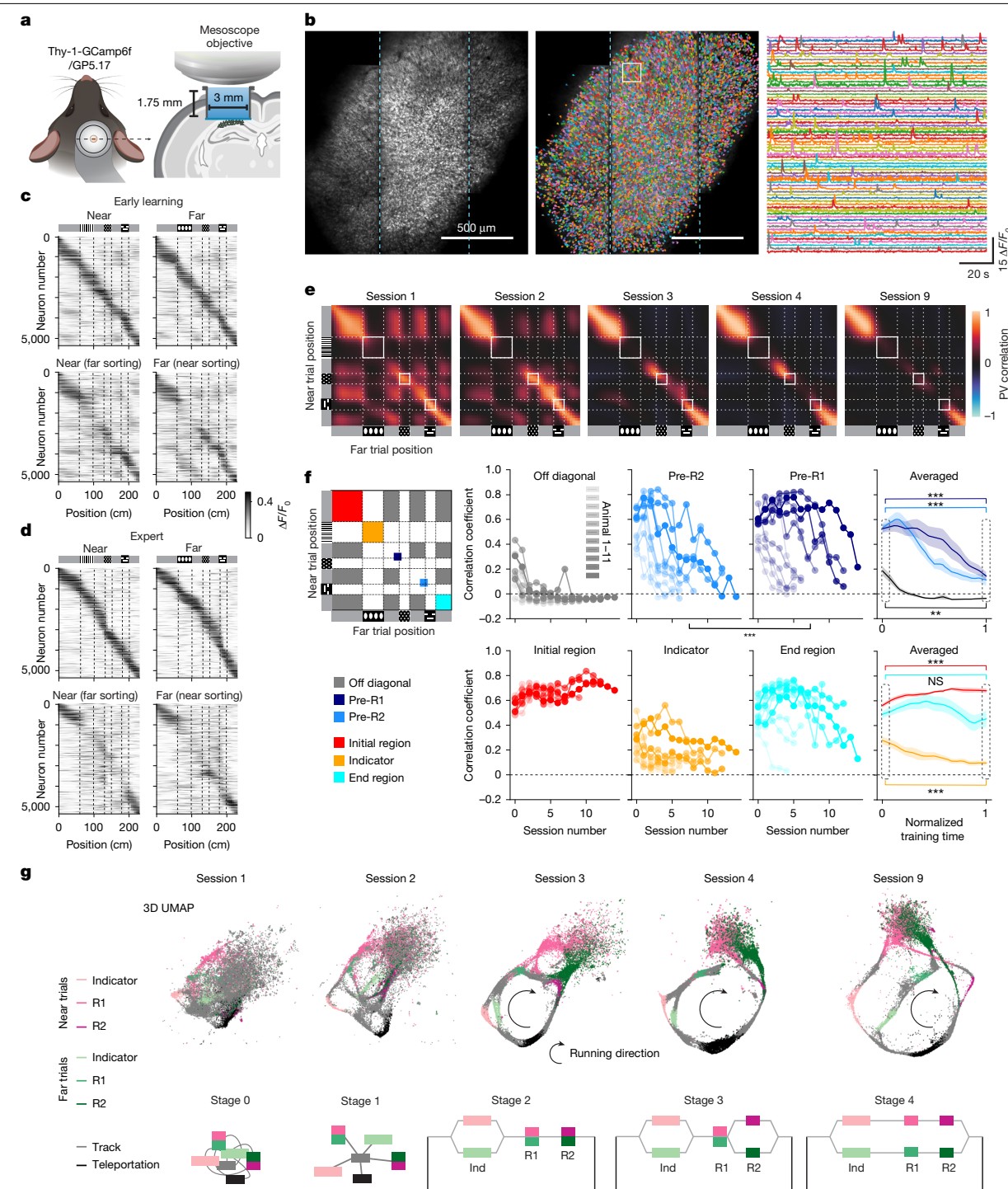

**Fig. 2 | Progressive decorrelation of neural activity during learning.**
**a**, Diagram of the CA1 imaging implant. The diagram of the CA1 imaging implant was created using BioRender (https://biorender.com). **b**, Example field of view, cell segmentations and extracted fluorescent signals. The cyan dashed lines mark the boundary between individual scanning stripes. Similar imaging and segmentation quality were observed across all 11 mice. **c**, Trial-averaged neural activity versus track position for both the near and the far trials on session 2 in a representative mouse (mouse 7 in Fig. 1f). Cells ordered by the centre of mass of the largest place field by indicated trial type (top) and cells ordered by the opposite trial type (bottom) are shown. Dashed lines mark the boundaries of indicator and reward cues. **d**, Similar to **c**, but for session 9 of the same mouse. **e**, Near-versus-far population vector (PV) cross-correlation matrices along track positions for sessions 1, 2, 3, 4 and 9 for the same example mouse. **f**, PV correlation averaged for different regions on the cross-correlation matrix across sessions, for off-diagonal grey region correlations (grey), pre-R2 region

(light blue), pre-R1 region (dark blue), initial region (red), indicator region (orange) and end region (cyan) shown for each mouse and averaged across all mice. Dashed lines in the matrix mark the boundaries of indicator and reward cues. The curves represent mean values, with shading indicating s.e.m. Comparing all sessions showed significant difference between pre-R1 and pre-R2 regions (two-sided Wilcoxon signed-rank test, ***$P = 9.96 \times 10^{-5}$, 97 total sessions from 11 mice). Comparisons between the first and the last session revealed significant changes in pre-R2, pre-R1, indicator (decreasing) and initial regions (increasing; ***$P < 0.001$; $n = 11$ mice). For the off-diagonal region, a significant decrease in correlation was observed (**$P = 0.002$; $n = 11$ mice). Changes in the end region were not significant (NS). **g**, 3D UMAPs and state diagrams of the neural manifold across learning for the five sessions and the same example mouse in **e**, with 2D views chosen to best illustrate learning dynamics captured by the 3D structure. Track diagrams for near and far trials (similar to Fig. 1a) are shown (**c**–**f**) for readability of the graphs.

animal-to-animal variability that occurs in both neural activity and behaviour suggests that some animals may learn differently (Fig. 1f and Extended Data Fig. 4).

## Hippocampal OSM representation

We further visualized the day-to-day dynamics of neural activity utilizing a non-linear dimensionality reduction technique, specifically uniform manifold approximation and projection (UMAP)[28]. A single embedding space was used to reduce the activities of thousands of cells to points in a low-dimensional (3D) UMAP space, using longitudinally registered data gathered across all days of imaging, in which each point represented the activity of all cells in a single imaging frame (Fig. 2g and Methods). Of note, this UMAP representation not only echoed the gradual decorrelation and orthogonalization that we previously described (Fig. 2e,f and Extended Data Figs. 4 and 5) but it also allowed us to intuitively observe the overall topological changes of the neural manifold during learning.

Here we describe the UMAP from a representative mouse exhibiting all learning stages. The UMAP representation from the initial session distinctly clustered the neural activity associated with each sensory cue. Despite this differentiation, the overall neural manifold appears relatively unstructured at this stage (Fig. 2g, stage 0). By the second day, the UMAP adopted a 'hub-and-spoke' appearance (Fig. 2g, stage 1), with the hub corresponding to all grey regions and the spokes corresponding to activity trajectories between a grey region and all other cues (that is, indicator cue, near and far reward cues, and the dark teleportation region). This structure hints that the neural activity associated with the concept of 'linear tracks' may not be fully developed at this point. An additional scattered point cloud near the reward-zone embeddings corresponded to periods of water-reward licks and a post-reward period during which the mouse was not running. By the third session, the UMAP adopted a ring-like structure that was closed by activity during the 2-s dark teleportation period linking the end of one trial and the start of the next trial (Fig. 2g, stage 2). As training progressed, the activity trajectories for the two trial types became increasingly distinct, eventually resembling a split-shank wedding ring, consisting of a band that splits into two strands with a diamond in the centre. Here, the splitting band corresponded to the principal manifold of neural activity while the mouse was running, and the diamond corresponded to the point cloud when the mouse was stationary, mostly during and right after reward consumption. We speculate that this reward-associated point cloud, observable in the UMAP representations at all stages, may be related to replay of neural activity and may contribute to synaptic plasticity in the hippocampus as well as its downstream targets[29] (Extended Data Fig. 7 and Supplementary Video 2 showing the single-trial UMAP dynamics). The gradual appearance of a split-shank ring UMAP mirrors the dynamics of the trial-type decorrelation described above (Fig. 2e,f).

The observed progressive changes in the representational structure reflected by both the correlation matrices and the UMAPs resemble a gradually evolving state machine undergoing several meaningful intermediate stages and finally reaching a structure capturing the essence of the task (Fig. 1a, bottom, and Fig. 2g, state diagrams below the UMAPs). This learning process involves several stages of disambiguation of similar sensory inputs at different regions along the two trial types in the populational activity level, eventually producing orthogonalized state representations for previously latent states of the task. Within this learned structure, the short-term memory of indicator cues is carried forwards by distinct neural activity representing different latent states. We call this learned representation of the task an OSM.

## Single-cell tuning changes during OSM formation

The orthogonalization of neural representations in the hippocampus reflects changes in the firing properties of individual neurons that occur alongside changes in the behaviour of an animal during learning. As training progresses, neurons undergo modifications in their tuning properties, becoming more selective and responsive to task-relevant features.

A prominent change observed in the early stages of learning is the transformation of neurons initially tuned to multiple grey regions into more selective cells, firing at fewer or even a single grey region (Fig. 3a). As learning continued, neurons displayed increasingly distinct tuning across near and far trial types, particularly in the pre-reward track regions (pre-R1 and pre-R2). This includes initially silent neurons that became active in specific regions for one trial type and not the other, as well as neurons that were initially active on both trial types but eventually became trial-type specific by decreasing activity in the other trial type (Fig. 3b,c). These 'splitter cells'[11,12] emerge throughout learning (Figs. 2e and 3).

To further characterize the diverse tuning properties of individual cells during the expert stage, for each cell, we calculated a difference score that quantifies the fractional difference in peak activity between the near and far trial types and plotted it against the correlation between the two responses (Fig. 3d). These two features separate the tuning responses of the cells into intuitive categories. Cells with large difference scores exhibit strong 'splitter' responses, showing large differences in tuning amplitude between trial types (for example, Fig. 3e, blue 1 and blue 2, and Fig. 3f, red 1 and red 2). Cells with low difference scores and high-correlation coefficients exhibit 'place' responses, which show similar tuning in both trial types (for example, Fig. 3e, blue 4). By contrast, cells with low difference scores and low-correlation coefficients exhibit 'remapping splitter' responses, whose tuning peaks have similar amplitudes but occur at different locations in near and far trials. Cells in the centre of the plot exhibit intermediate phenotypes. For example, some cells show a combination of place and splitter phenotypes (for example, Fig. 3f, red 4). The cells in the centre with moderate values for both features highlight that the distinction between place cells and splitter cells are best described as a continuum of responses with diverse tuning properties. Furthermore, these response properties are plastic, and learning probably reflects these changing neuronal responses.

We examined the emergence of these response types over learning by separating cells based on their maximum tuning locations: trial start and end, indicator region, and reward or pre-reward regions (Fig. 3h, left). Plotting their positions in the difference score versus correlation scatter plot for novice, intermediate and expert sessions revealed gradual changes across learning for each group. Track start or track end responses were initially concentrated near the moderately high correlation and low-to-intermediate difference score region, suggesting variable but mostly correlated tuning. With learning, these cells quickly adopted more obvious place responses with very high correlations and low difference scores, consistent with the population correlation analysis showing high correlation at the trial start or trial end even in well-trained mice (Fig. 2e–g). Indicator-tuned cells transitioned from a scattered distribution to a concentrated density in the upper region with high difference scores, highlighting their rapid transformation into responses that more completely distinguished between different visual cues. By contrast, cells tuned to the reward and pre-reward regions gradually transitioned from place-like responses to splitter responses, indicating that these sensory-ambiguous regions require more prolonged learning to produce differential responses.

Quantifying the percentage of cells with responses in three arbitrarily defined regions of the scatter plot revealed a gradual increase in the percentage of splitter responses and a corresponding drop in the place-like responses during learning, whereas remapping splitter responses remained low (Fig. 3i). Although for simplicity we quantified responses in these categories, in reality they represent points on a continuum rather than discrete cell types. At the expert stage, cells with place and place-splitter responses dominated the track start and end

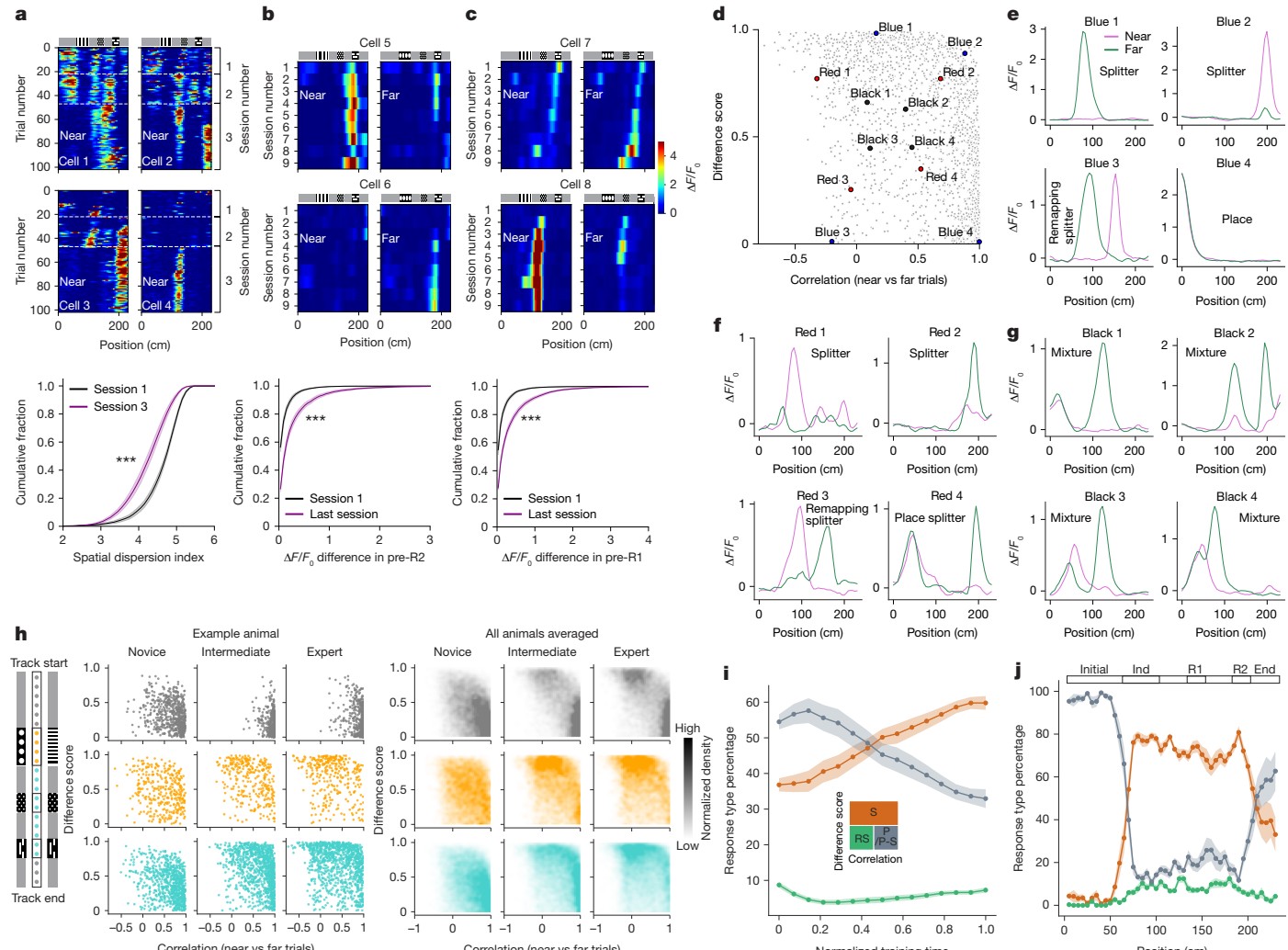

**Fig. 3 | Single-cell tuning changes during learning. a**, Positional tuning for four example cells tuned to grey regions across near trials during stage 1–2 transition (first three sessions; top), and the cumulative percentage of the spatial dispersion index for grey region-tuned cells in near trials, session 1 versus session 3 (bottom; plot showing mean ± s.e.m., two-sided Wilcoxon rank-sum test, ***$P < 1 × 10^{-6}$; $n = 11$ mice). **b**, Session-averaged activity for near or far trials across nine sessions for two pre-R2-tuned cells (top), and the cumulative percentage of $\Delta F/F_0$ difference in pre-R2 between trial types for tuned cells, sessions 1 versus 9 (bottom; plot showing mean ± s.e.m., two-sided Wilcoxon rank-sum test, ***$P < 1 × 10^{-6}$; $n = 11$ mice). **c**, Similar to **b** for the pre-R1 region (plot showing mean ± s.e.m., ***$P < 1 × 10^{-6}$; $n = 11$ mice). **d**, Correlation coefficient versus difference score scatter plot for cells with activity of more than 2 s.d. above the mean in the expert session. The *x* axis indicates the near–far correlation coefficient, and the *y* axis denotes the difference score $D = |A\_near − A\_far|/\max(A\_near, A\_far)$, in which $A\_near$ and $A\_far$ are peak neural

activities in near and far trials. Example cells were manually selected and colour-coded based on their relative location within the plot. **e**–**g**, Trial-averaged tuning curves for near (magenta) and far (green) trials for example cells from **d**, labelled by tuning phenotype. **h**, Scatter plots (left) showing learning progression (novice, intermediate and expert) for cells with maximum tuning in the initial or end regions (grey), indicator region (red), and reward or pre-reward regions (cyan), and normalized density plots (right) across mice for each track-segment-tuned population. **i**, Percentage of place cell and place-splitter cell (P/P-S), splitter cell (S) and remapping splitter cell (RS) responses across training (categorized by correlation = 0.2, difference score = 0.5 thresholds; mean ± s.e.m.; $n = 11$ mice). **j**, Distribution of response types along track regions at the expert stage (mean ± s.e.m.; $n = 11$ mice). Track diagrams for near and far trials (similar to Fig. 1a) are shown (**a**–**c**,**h**) for readability of the graphs.

regions, whereas cells with splitter and remapping splitter responses dominated the regions in the middle of the track (Fig. 3j). In summary, these single-cell tuning changes can be understood as the hippocampus learning to extract the latent task structure despite the ambiguity of immediate sensory experiences. To facilitate exploration of these diverse single-cell tuning properties, we developed an interactive visualization tool, which is available at http://cognitivemap.janelia.org/.

## Hippocampal maps versus computational models

The large number of neurons that we recorded over many days of training presents a unique opportunity to probe the learning algorithms

that lead to the gradual emergence and final representational structure of the 2ACDC task. Several recent theoretical models have conceptualized cognitive maps as learned internal models of the world that allow animals to predict upcoming sensory experiences from their understanding of the environment and their actions in it[20,21]. To test whether this class of models can provide insight into the measured hippocampal learning, we analysed an HMM-based model called the CSCG[21,23]. Fundamentally, HMMs and CSCGs aim to uncover hidden structures from sequential data, capturing meaningful latent states and their temporal dependencies (Fig. 4a). CSCGs make use of 'clones' that assign states to fixed sensory observations via a deterministic emission matrix (Fig. 4a,b). State occupancy probabilities are influenced

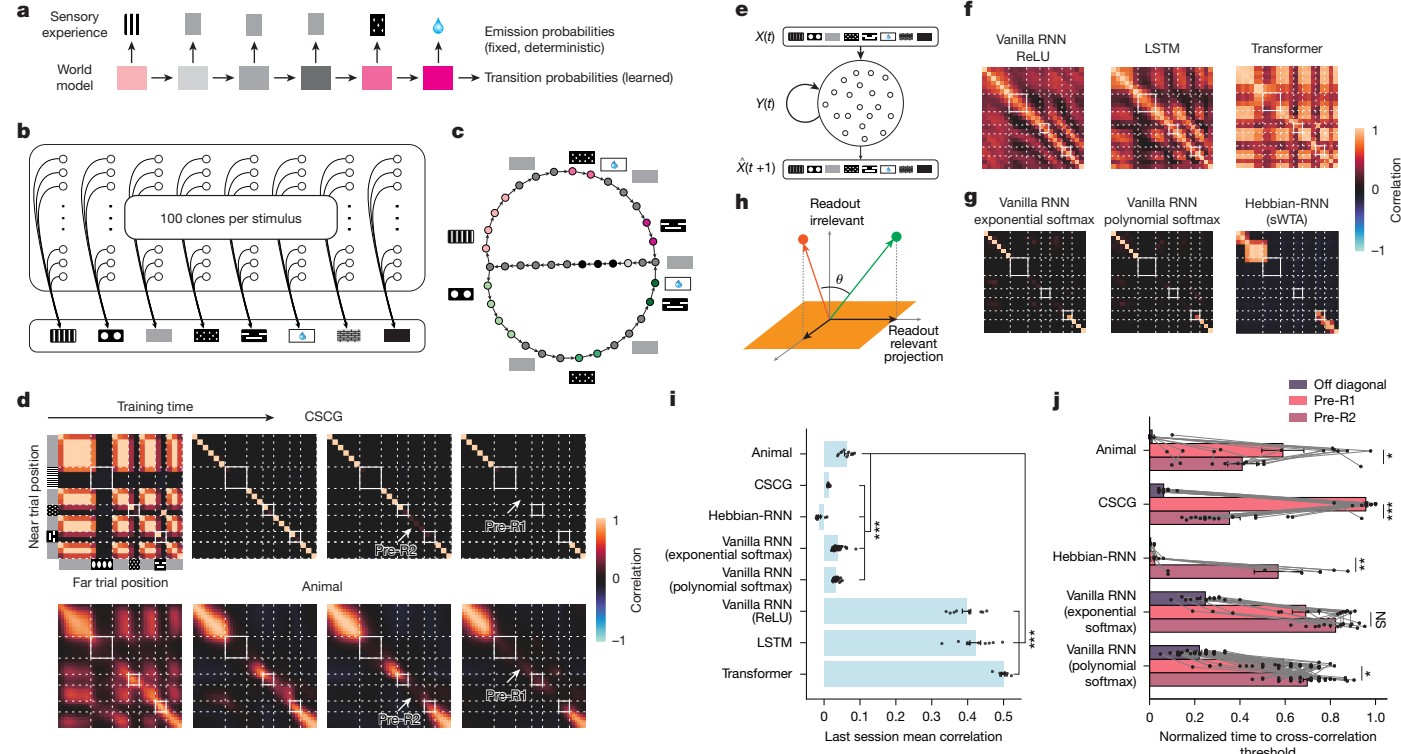

**Fig. 4 | Representational structure during learning for mice and different models. a**, HMM diagram showing latent states and transition probabilities with fixed emission probabilities. **b**, CSCG diagram with 100 clones per sensory symbol. **c**, Final transition graph of the CSCG trained on the 2ACDC task, showing distinct latent states and their sensory inputs. Coloured circles indicate specific track regions similar to Fig. 2g. **d**, CSCG (top) and CA1 (bottom) cross-correlation matrices between near and far trials during learning (the CA1 data are the same as Fig. 2e; sessions 1, 3, 4 and 9). Track diagrams for near and far trials (similar to Fig. 1a) are shown for readability of the graphs. **e**, RNN diagram showing sensory input processing ($X(t)$) through recurrent units, which produce the hidden state $Y(t)$, to predict next input ($\hat{X}(t+1)$). **f,g**, Final correlation matrices for near versus far trial representations across models. Polynomial softmax uses 8th power normalization. **h**, Schematic showing how correlated high-dimensional activity (green and red vectors) can yield orthogonal outputs through readout projection (yellow plane). **i**, Mean final correlation matrix quantification across mice ($n = 11$) and models (independent simulations runs: $n = 20$ (CSCG), $n = 10$ (Hebbian-RNN), $n = 42$ (vanilla RNN exponential softmax), $n = 47$ (polynomial softmax), $n = 10$ (rectified linear (ReLU)), $n = 10$ (LSTM) and $n = 10$ (transformer); two-sided unpaired Student's $t$-tests with Bonferroni correction for multiple comparisons). **j**, Time for key regions to reach correlation threshold (0.3), normalized to training duration. Data from model runs ($n = 18$ (CSCG), $n = 8$ (Hebbian-RNN), $n = 15$ (vanilla RNN exponential softmax) and $n = 39$ (polynomial softmax)) and mice ($n = 11$ mice; two-sided paired Student's $t$-tests without correction for multiple comparisons). The bar graphs in **i,j** show mean ± s.e.m.; *$P < 0.05$, **$P < 0.01$ and ***$P < 0.001$.

by current and past sensory stimuli, and the model was constructed by finding a transition matrix between states that best predicts sensory sequences using the Baum–Welch expectation maximization algorithm[30] (Fig. 4a, Extended Data Fig. 8 and Methods).

The CSCG model recapitulated the final representation and learning trajectory of the hippocampus. Like the UMAP embedding of hippocampal activity (Fig. 2g), the learned state transitions of the CSCG closely resembled the task architecture (Fig. 4c), with trial-type-specific state sequences producing distinct sensory experiences on the two trial types (Fig. 4c, upper and lower branches), and shared state sequences producing the sensory sequence shared across trials (Fig. 4c, middle branch). We then compared clone occupancy probabilities to population vector activity in the hippocampus. We found that during training, the state probabilities gradually progressed through stages of orthogonalization that closely mirrored the dynamics that we observed in the hippocampus (Fig. 4d and Extended Data Fig. 8). The ability of the CSCG model to recapitulate key features of hippocampal neural dynamics suggests that sequence-dependent extraction of latent states from potentially ambiguous or identical sensory inputs may represent a fundamental computational principle of hippocampal learning.

These successes of CSCG are not necessary for task performance. For example, we investigated whether RNNs would produce orthogonalized representations when trained to predict the next sensory input

in the 2ACDC task sequence (Fig. 4e). RNNs using rectified linear or sigmoid activation functions trained using backpropagation through time achieved high accuracy in predicting next sensory inputs without developing the orthogonalized representations characteristic of hippocampal activity (Fig. 4f and Extended Data Fig. 9). In these models, activity corresponding to the same sensory inputs in different latent states remained highly correlated, contrasting sharply with our experimental observations in mice (Fig. 4f,i). This is because perfect task performance only requires that population neural activity be orthogonal in the low-dimensional task-relevant subspace that is read out for stimulus prediction, leaving many task-irrelevant dimensions that have no effect on task performance (Fig. 4h). Similarly, more complex neural network architectures widely used in sequence learning tasks, LSTM[24] networks and transformers[25] achieved high prediction accuracy but did not naturally produce orthogonalized representations unless explicitly encouraged to learn the orthogonalized representations as part of their learning objective (Fig. 4f,i and Extended Data Fig. 9). Conversely, RNNs with softmax activation functions did produce fully orthogonalized representations when fully trained, more closely resembling the hippocampal data (Fig. 4g,i). This shows that in addition to learning rules, the learning objective and the choice of architectural features critically influences the final representational structure of these networks.

Biologically plausible neural network models and plasticity rules can also produce hippocampus-like representations. Previous work has suggested that spike-timing-dependent plasticity[31,32] can stably encode sequences[33] in a manner that is robust to noise[34]. Spike-timing-dependent plasticity has also been shown theoretically to facilitate forming predictive maps[35–37] and approximate HMM learning[38]. We thus built a spiking RNN model that included a soft winner-take-all (sWTA) mechanism, which leverages the principle of feedback inhibition to ensure that only the highest firing neurons remain active within the network. Using only a timing-based Hebbian plasticity rule based on local activity[38] (that is, no end-to-end training or explicit task), the model (Hebbian-RNN) learned orthogonalized representations of the 2ACDC task (Fig. 4g and Extended Data Fig. 10). These findings underscore the ability of canonical, biologically plausible learning mechanisms to shape hippocampal representations and suggest that sWTA-like mechanisms help to promote decorrelated cognitive maps. Although the correlation values of the mice were slightly higher than those of the fully decorrelated models (Fig. 4i), this difference may be attributed to ongoing learning processes in the mice at the time of measurement. In rapidly decorrelating regions such as the off-diagonal areas, mice showed near-complete decorrelation (Fig. 2f).

Crucially, the specific decorrelation sequence observed during learning provided a stringent constraint on potential models of hippocampal function. In our experimental data, we observed an average pattern in which off-diagonal elements decorrelated first, followed by the pre-R2 region, and finally the pre-R1 region (Fig. 2e,f). Among the models tested, only the CSCG consistently reproduced this precise decorrelation trajectory (Fig. 4d,j). Other models that achieved decorrelated final states, including vanilla RNNs and Hebbian-RNNs, showed different sequences of decorrelation, with pre-R1 often decorrelating before or simultaneously with pre-R2 (Fig. 4j). This distinction in learning dynamics provides a critical means of discriminating between potential algorithmic accounts of hippocampal function. It also suggests that the CSCG based on the Baum–Welch expectation maximization algorithm captures critical algorithmic properties of hippocampal learning that can inform future work to mechanistically explain cognitive map formation through biologically plausible plasticity rules. Although these results support the CSCG as a leading model, further research is needed to fully elucidate the complex mechanisms and principles contributing to cognitive map formation.

## Adaptation of the existing hippocampal state machine

To investigate whether and how the learned hippocampal state machine would adapt to novel task features, we expanded and modified the structure of the task. First, after mice learned the task with the original indicator cues (cue pair A), we replaced them with two unfamiliar visual patterns. To do this, we developed four unique indicator pairs (cue pairs B, C, D and E) and presented them to mice that had already learned the original cue pair (Fig. 5a). Every day, the mice were initially exposed to cue pair A for a duration of 5–10 min, after which the indicators for the task were replaced with one of the novel pairs. This change enabled us to collect neural activity data for both the original and the new cue pairs during the same session. Training on the new cue pair continued until the mouse could proficiently execute the task, demonstrated by restricting its licking to the rewarded location or just before it on 75% of the trials for three successive sessions. Mice were then sequentially trained on each subsequent novel indicator pair on the following days in the same manner. Through this training process, mice learned the new cue pairs in significantly fewer trials (147 ± 39 trials for the new cue pairs compared with 483 ± 70 trials for the original cue pair; $n = 3$ mice; *$P < 0.05$, unpaired Student's $t$-test; Fig. 5b).

Comparing population vector correlations of the neural activity for trials with novel indicators versus the original indicators revealed high similarities in neural representations between the old and the new tasks in all track regions except the indicator region (Fig. 5c). In other words, the neural activity in the presence of new indicators mirrored the common task structure while maintaining information about the visual identities of the novel indicator cues. In terms of the state machine framework, this suggests that once a state machine is established in the hippocampus, it can be effectively reused for new task variants. New task elements can be incorporated into the existing state machine, either through the creation of new states or linking new sensory inputs to existing states (Fig. 5d). This flexible adaptation and integration, in turn, expedite learning.

In a second variation of the task, we extended the length of the grey zones following the indicator cue and after the first reward cue, thus requiring animals to travel longer distances to reach the reward zones (Fig. 5e,f and Methods). We inserted these 'stretched trials' after every 5–6 regular trials to evaluate how well-trained mice for the original task respond to the altered environment without extensive adjustment to these task modifications (Methods). In both stretched near (near') and stretched far (far') trial types, mice displayed a tendency to lick towards the beginning of the usual reward location, even though the reward cue was not yet encountered (Extended Data Fig. 9). A comparison of the tuning location for each cell during normal and stretched trials of the same type (that is, near–near' or far–far') provided insights into how the mice might perceive the modified task (Fig. 5e,f). As expected, place fields were tuned to similar locations in both regular and stretched trials for the initial, unmodified portion of the track. When mice entered the first stretched region in near trials, cells normally tuned to the grey region before the first reward (pre-R1) maintained their activity throughout the extended segment (Fig. 5e; $n = 3$ mice). This potentially indicates that the animal may believe it remains in the same patch of the grey zone right before the first reward region. During far trials, the difference in reward location expectation produces a different pattern. When mice entered the first stretched region in far trials, neurons did not persistently extend their activity. Instead, they rapidly shifted their tuning to align with the region just before the second reward area (pre-R2), as if the mouse was anticipating the second reward location (Fig. 5f). However, when the mouse eventually saw the first reward cue, the representation quickly resets and anchors to the representation reflecting the first reward region. These results imply that, when the animals encounter modified components of the task, the neural representations can settle into discrete states to mirror inferred latent states under conditions of uncertainty (Fig. 5e,f). These discoveries substantiate the idea that the learned cognitive map exhibits the properties of a state machine that can infer and flexibly use learned states in novel situations. Investigating how various computational models respond to such task alterations presents an exciting avenue for future research[39].

## Discussion

We assessed neural activity in a large population of neurons as mice learned a virtual reality task over the course of several days to weeks. Our findings reveal the gradual emergence of a cognitive map within the hippocampus, coinciding with improvements in task performance. This development is reflected in changes to both population-level neural activity and the response properties of individual neurons. Well-trained mice exhibit robust short-term and long-term memory – processes that are consistent with the structure of the mature cognitive map, including the ability to produce effective behaviour in novel environments with similar structure but altered features. The cognitive map has features of a state machine with orthogonalized representations of latent states that the animal must discover to perform the task efficiently and that meaningfully represent altered versions of the environment. Computational modelling suggests that many features of this OSM, including

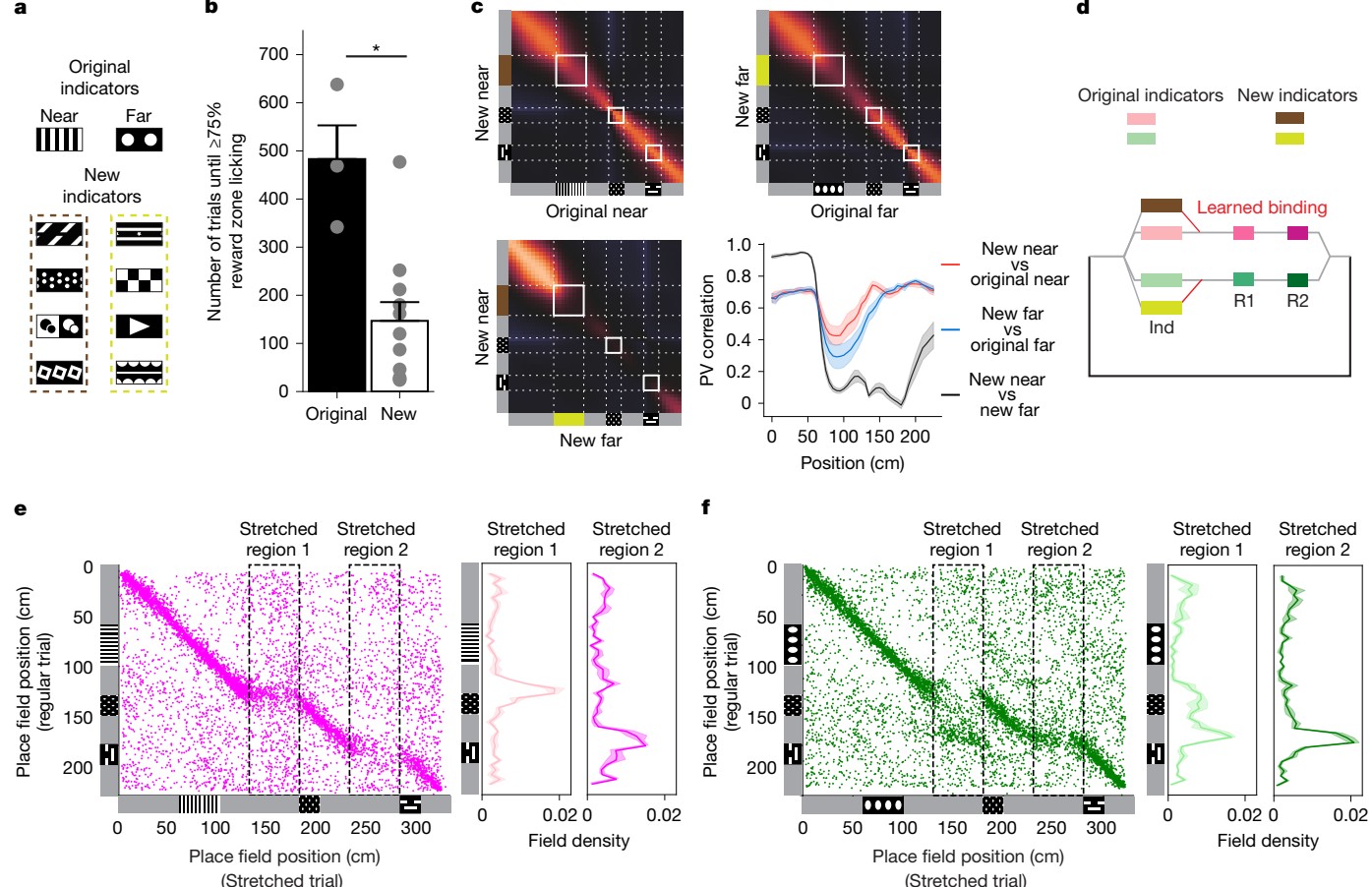

**Fig. 5 | State machines can be flexibly used in novel settings. a**, Original and novel indicator pairs. **b**, Number of trials required to reach high performance (75% or more correct licking) for each indicator pair. For original indicators, $n = 3$ mice; for novel indicators, $n = 11$ training periods across the same 3 mice; *$P = 0.039$, two-sided unpaired Student's $t$-test. The bar graphs show mean ± s.e.m.; the dots represent individual training periods. **c**, PV cross-correlation between the neural activity for trials with the original and new indicators for the same trial type (near (top left) and far (top right)), and between near and far trials for the new indicators (bottom left). Quantification of the diagonal cross-correlations in three cross-correlation matrices (line and shading indicate mean ± s.e.m.; $n = 3$ mice) is also shown (bottom right). **d**, Conceptual diagram for the incorporation of new indicator states into an existing state machine. **e**, Place field locations in the stretched near trials plotted against those in the regular near trials ($n = 3$ mice, data pooled together). The histograms of field locations in the two stretched regions are plotted to the right (line and shading indicate mean ± s.e.m.; $n = 3$ mice). **f**, Similar to **e** but for the far trials. Track diagrams for near and far trials (similar to Fig. 1a) have been added to **c**,**e**,**f**.

the gradual dynamics of its formation, share properties with a type of HMM called CSCG that is learned using the expectation maximization algorithm.

We further showed that RNNs can also produce orthogonalized representations under certain conditions. Vanilla RNNs trained with backpropagation through time can achieve orthogonalized representations when using sWTA activation functions such as softmax. In addition, biologically plausible spiking RNNs coupled with sWTA dynamics and trained using Hebbian plasticity are sufficient to construct the final OSM. This combination of RNN, sWTA and Hebbian plasticity has been previously shown to approximate HMM learning via expectation maximization[38], highlighting that known biological mechanisms may construct graph-like representations of environments where animals repeatedly experience sequences of sensory stimuli, including rewards that are delivered in latent contexts. However, although these RNN models capture the final orthogonalized representations, the specific sequences of decorrelation do not match those observed in animals. In addition, we found that widely used sequence learning models in artificial intelligence, specifically LSTMs and transformers, trained to predict the next element in a sequence using backpropagation of error, do not naturally produce orthogonalized representations like those found in the mouse hippocampus. However, we found that this

key property of the hippocampal OSM could be observed in LSTM when the cost function explicitly penalized activity correlation between the two trial types. Our modelling results suggest that the emergence of orthogonalized representations could be driven by specific architectural choices (such as activation functions) and cost functions rather than the learning algorithm alone. Our approach can be viewed as a feature-matching method for inferring learning rules from neural activity dynamics, complementing recent work on identifying learning rules from neural observables[40,41].

CSCG captures both the final representations and the learning trajectory observed in animals, strengthening its promise for modelling hippocampal function[21,23]. The success of the CSCG compared with RNNs may be attributed to its training via Baum–Welch expectation maximization, its ability to decouple latent dynamics from observations and its use of discrete representations. These features make the CSCG particularly well suited for planning and reasoning in artificial intelligence systems, mirroring the role of the hippocampus in flexible behaviour.

These results have important implications for understanding the architectural features, objectives and plasticity mechanisms that may contribute to the formation of the hippocampal cognitive maps. Foremost among them, our modelling indicates that both local Hebbian

learning and gradient descent learning with appropriate architectural choices are sufficient to construct the orthogonalized map. Hebbian plasticity does not require feedback from other brain areas, as it is determined entirely by the relative timing of presynaptic and postsynaptic spikes local to the modified synapse[32]. This is fundamentally different from the methods typically used to adjust synaptic weights in artificial neural networks, such as in vanilla RNNs, LSTMs and transformers, which rely on backpropagation of error through explicitly defined cost functions[42]. Nevertheless, there is good evidence that feedback-based plasticity is important in the hippocampus[43,44], and it has been proposed to be a key element in approximating the backpropagation of error algorithm in the brain[45]. Feedback-based mechanisms involving target, error or reward signals are also likely to be instrumental. These mechanisms may work in tandem with Hebbian plasticity to construct cognitive maps and/or they may be more involved in refining behavioural policy and other task-specific functions by selectively routing information from the established cognitive maps to other brain regions mediating behavioural policies. Our data indicate that task representations and behavioural policies based on them are formed in lockstep, as previously suggested[46]. A likely candidate mechanism for the contribution of synaptic plasticity during feedback is behavioural timescale synaptic plasticity[43]. Future experiments replacing rewards with novel sensory cues could clarify whether reward is necessary for extracting latent task structure or whether sensory prediction alone suffices. Such studies would distinguish between reward-driven learning and purely sensory-based predictive coding in cognitive map formation. In addition to these biological mechanisms, recent machine learning advances offer new perspectives on achieving decorrelation through backpropagation-like processes. Techniques such as contrastive losses[47] and object-centric representations[48] provide alternative approaches to generating decorrelated representations, which may have parallels in biological learning systems.

It is important to note that plasticity mechanisms in other brain regions probably contribute to our observations. For example, CA1 receives most of its excitatory synaptic input from CA3, where recurrent connections[49], attractor dynamics[50] and Hebbian plasticity have all been observed[51]. Plasticity in CA3, as well as other brain regions, may thus result in changes in the firing of the pyramidal neurons that we imaged in CA1. Although our RNN models are not detailed representations of hippocampal circuitry, they can be considered as abstractions of CA3 function. Although we recorded from CA1, it is plausible that CA1 inherits orthogonalized representations from upstream regions such as CA3. We propose that the existence of multiple forms of synaptic plasticity across different brain regions allows unsupervised and supervised (and reinforcement) learning to work together to reduce sensory interference, build robust models of the environment and direct the content of these models to promote adaptive behaviours. Understanding how these diverse plasticity mechanisms interact to produce flexible and efficient cognitive maps remains a key challenge in understanding the computational principles of the brain. Future research should focus on elucidating the specific roles and interactions of Hebbian, feedback-based and other learning rules in cognitive map formation and utilization across different brain regions. Identifying the loci and molecular mechanisms of these processes will be crucial for advancing our understanding of how the brain learns and adapts to complex environments.

A classical concept in computer science, a finite state machine is a computational structure consisting of a finite set of states with the transitions between them based on defined inputs or conditions[52]. States reflect current sensory input from the environment and the body of the animal, as well as latent information such as the recent history of sequential observations. Transitions are constrained by the current state and neurally encoded transition probabilities, and determined by the movements of the animal and the sensory input it receives from the environment. We posit that neural activity in the hippocampal OSM

could contribute to adaptive behaviour, such as speeding up, slowing down or licking[53]. These behaviours in turn influence neural activity, and thus transitions to new states in the hippocampal OSM, both by changing the external and internal sensory experience of the animal and by changing the stimuli coming from the environment. The hippocampal OSM operates in closed loop with the rest of the brain, the body of the animal and its environment to produce the properties of a state machine.

The emergence of the OSM involves dynamic changes in single-cell tuning properties, which we characterized using a novel 2D feature space. This approach reveals a continuum of response types rather than discrete categories, challenging traditional cell-type classifications and demonstrating that single-cell tuning, when properly interpreted in the context of a latent state inference[20,21,23,54], can provide valuable insights into learning dynamics at a fine granularity. We observed that individual neurons dynamically transition between functional roles as learning progresses, adapting their representations to capture task-relevant information. This flexibility suggests that hippocampal neurons act more like plastic state cells than rigid place or splitter cells. Plasticity of these features is responsible for the gradual discovery and representation of task-related latent states. Our findings extend beyond the concepts of splitter cells and pattern separation, showing that cognitive map formation is not a simple accumulation of these phenomena but a systematic, stereotypical progression in neural representations. Our findings support the long-standing proposal that sparse orthogonal representations are a powerful mechanism for memory and intelligence[55,56].

Our modelling efforts complement existing frameworks in hippocampal research by specifically addressing the dynamics of learning in ambiguous environments such as the 2ACDC task that we used here. We explored a range of models to capture the gradual emergence of orthogonalized representations observed in our data. This approach revealed that certain computational principles, particularly those involving latent state inference, are crucial for replicating both the final representations and the learning trajectory seen in animals. Although influential models such as successor representations[19,57] and the Tolman–Eichenbaum machine[20] have provided valuable insights into cognitive maps, they focus on different aspects of hippocampal function. Successor representations efficiently compute long-term reward predictions by factoring the value function into state dynamics and reward but require pre-defined states. The Tolman–Eichenbaum machine emphasizes generalization of structural knowledge across environments and does not address how new latent states could be rapidly learned in new environments. Our work, similar in conceptualization to the CSCG[21,23], extends the field by providing data elucidating how the hippocampus learns de novo latent states and resolves perceptual ambiguity within a single environment, a process fundamental to the formation of precise cognitive maps.

Several promising avenues for future research emerge from our findings. Although our models capture key aspects of hippocampal representations, further work is needed to fully explain the specific sequence of orthogonalization observed in animals. Future models could incorporate additional biological mechanisms, such as replay[29], to more accurately mirror the learning process. In addition, investigating the interaction between hippocampal and neocortical representations[58] during learning could provide valuable insights into how the brain forms and uses cognitive maps across different timescales and levels of abstraction.

In conclusion, our study provides a comprehensive view of cognitive map formation in the hippocampus during learning of a moderately complex task. The emergence of OSM representations offers a fundamental framework for understanding neural computation, memory and intelligence. Our findings reveal key principles underlying biological cognitive map formation, offering insights that could both deepen our understanding of adaptive behaviour and guide the

development of more sophisticated artificial intelligence systems with robust world models.

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

## Methods

All procedures were performed in accordance with the Janelia Research Campus Institutional Animal Care and Use Committee guidelines. Both male and female GCaMP6f (Thy1-GCaMP6f[59]) transgenic mice were used, 3–6 months of age at the time of surgery (3–8 months of age at the beginning of imaging studies).

### Surgery

Mice were anaesthetized with 1.5–2.0% isoflurane. A craniotomy on the right hemisphere was performed, centred at 1.8 mm anteroposterior and 2.0 mm mediolateral from the bregma using a 3-mm diameter trephine drill bit. The overlying cortex of the dorsal hippocampus was then gently aspirated with a 25-gauge blunt-tip needle under cold saline. A 3-mm glass coverslip previously attached to a stainless-steel cannula using optical glue was implanted over the dorsal CA1 region. The upper part of the cannula and a custom titanium headbar were finally secured to the skull with dental cement. Mice were allowed to recover for a minimum of 2 days before being put under water restriction (1.0–1.5 ml daily), in a reversed dark–light cycle room (12-h light–dark cycle).

### Behaviour

**Virtual reality setup.** The virtual reality behavioural setup was based on a design previously described[60]. The spherical treadmill consisted of a hollowed-out Styrofoam ball (diameter of 16 inches, 65 g) air-suspended on a bed of 10 air-cushioned ping-pong balls in an acrylic frame. Mice were head fixed on top of the treadmill using a motorized holder (Zaber T-RSW60A; MOG-130-10 and MOZ-200-25, Optics Focus) with their eyes approximately 20 mm above the surface. To translate the movement of the treadmill into virtual reality, two cameras separated at 90° were focused on 4-mm$^2$ regions of the equator of the ball under infrared light[60]. Three axis movement of the ball was captured by comparing the movement between consecutive frames at 4 kHz and readout at 200 Hz (ref. 60). A stainless-steel tube (inner diameter of 0.046 inches), attached to a three-axis motorized stage assembly (Zaber NA11B30-T4A-MC03, TSB28E14, LSA25A-T4A and X-MCB2-KX15B), was positioned in front of the mouse's mouth for delivery of water rewards. The mouse was shown a perspective corrected view of the virtual reality environment through three screens (LG LP097QX1 with Adafruit Qualia bare driver board) placed roughly 13 cm away from the animal (Fig. 1a). This screen assembly could be swivelled into position using a fixed support beam. All rendering, task logic and logging were handled by a custom software package called Gimbl (https://github.com/winnubstj/Gimbl) for the Unity game engine (https://unity.com/). All inter-device communication was handled by a MQTT messaging broker hosted on the virtual reality computer. Synchronization of the virtual reality state with the calcium imaging was achieved by overlaying the frame trigger signal of the microscope with timing information from inbuild Unity frame event functions.

To observe the mouse during the task without blocking its field of view, we integrated a periscope design into the monitor assembly. Crucially, this included a 45° hot-mirror mounted at the base of a side monitor that passed through visible light but reflected infrared light (Edmund Optics 62-630). A camera (Flea3-FL3-U3-13Y3M) aimed at a secondary mirror on top of the monitor assembly could hereby image a clear side view of the face of the mouse. Using this camera, a custom Bonsai script[61] monitored the area around the tip of the lick port and detected licks of the mouse in real time that were used in the virtual reality task as described below.

**Head fixation training.** After recovering from surgery, mice were placed on water restriction (1.0–1.5 ml daily) for at least 2 weeks before behavioural training. Body weight and overall health indicators were checked every day to ensure mice remained healthy during the training process. Mice were acclimated to experimenter handling for

3 days by hand delivering water using a syringe. For the next three sessions, with the virtual reality screens turned off, mice were head fixed on the spherical treadmill while water was randomly dispensed from the lick port (10 ± 3 s interval; 5 µl per reward). These sessions lasted until mice acquired their daily allotment of water or until 1 h had passed. We observed that during this period most mice started to run on their own volition. Next, we linked water rewards to bouts of persistent running and increased this duration across sessions till the mouse would run for at least 2 s straight (approximately five sessions). During this time, we also slowly increased the height of the animal with respect to the treadmill surface across sessions to improve performance. Mice that did not show sufficient running behaviour to acquire their daily allotment of water were discarded in further experiments.

**2ACDC task.** At the beginning of each trial, mice were placed at the start of a virtual 230-cm corridor. The appearance of the walls was uniform except at the location of three visual cues that represented the indicator cue (40 cm long) and the two reward cues (near or far; 20 cm long). Depending on the trial type, a water reward (5 µl) could be obtained at either the near or far reward cue (near and far reward trials). The only visual signifier for the current trial type was the identity of the indicator cue at the start of the corridor. For the first 2–3 sessions, the mouse only had to run past the correct reward cue to trigger reward delivery ('guided' sessions). On all subsequent sessions, mice had to lick at the correct reward cue ('operant' sessions). No penalty was given for licking at the incorrect reward cue. In other words, if the mouse licked at the near reward cue during a far trial type, then the mouse could still receive a reward at the later far reward cue. Upon reaching the end of the corridor, the virtual reality screen would slowly dim to black, and mice would be teleported to the start of the corridor to begin the next semi-randomly chosen trial with a 2-s duration. The probability of each trial type was 50%, but to prevent bias formation caused by very long stretches of the same trial type, sets of near or far trials were interleaved with their number of repeats set by a random limited Poisson sampling (lambda = 0.7, max repeats = 3). The identity of the indicator cue was kept hidden for the first 20 cm of the trial and was rendered when the mice passed the 20-cm position. To internally track the learning progress of the mouse, we utilized a binarized accuracy score for each trial depending on whether the mouse only licked at the correct reward cue. Once the mouse had three sessions in which the average accuracy was above 75%, we considered the mouse to have learned that cue pair.

**2ACDC task with novel indicators.** For 3 mice out of the 11 well-trained mice on the original 2ACDC task, we subsequently trained them to perform the 2ACDC task novel indicator pairs. After reaching three consecutive sessions with more than 75% task accuracy for the original 2ACDC task, the novel task was introduced in the following session, but with the original task shown for the first 5–10 min at the beginning of each session before switching completely to the new task. When the mouse could perform the new task for 3 consecutive days with more than 75% accuracy, we moved on to the next novel indicator pair until the last one was finished (four novel indicator pairs in total).

**2ACDC task with extended grey regions.** As another modification to the original task design, the grey regions were extended in certain trials, which we called the 'stretched trials'. In the stretched trials, the linear track was extended from 230 cm to 330 cm, and the reward positions were moved from [130, 150] cm to [180, 200] cm (the first rewarding (near) object), and [180, 200] cm to [280, 300] cm (the second rewarding (far) object). Note that the distance between the indicator cue and the near object in the stretch trial is equal to the one between the indicator cues and the far object in the normal trial. During a session with stretch trials, following a 5-min warm-up using only the normal 2ACDC trial, the stretch trial was adopted at intervals of every five or six trials.

**Calcium imaging.** Neural activity was recorded using a custom-made two-photon random access (2P-RAM) mesoscope[27] and data acquired through ScanImage software[62], running on MATLAB 2021a. GCaMP6f was excited at 920 nm (Chameleon Ultra II, Coherent). Three adjacent regions of interest (each 650 μm wide) were used to image dorsal CA1 neurons. The size of the regions of interest was adjusted to ensure a scanning frequency at 10 Hz. Calcium imaging data were saved into tiff files and were processed using the Suite2p toolbox (https://www.suite2p.org/). This included motion correction, cell regions of interest, neuropil correction and spike deconvolution as described elsewhere[63].

**Multiday alignment.** To image the same cells across subsequent days, we utilized a combination of mechanical, optical and computational alignment steps (Extended Data Figs. 2 and 3). First, mice were head fixed using a motorized head bar holder (see above), allowing precise control along three axes (roll, pitch and height) with submicron precision. Coordinates were carefully chosen at the start of training to allow for unimpeded animal movement and reused across subsequent sessions. The 2P-RAM microscope was mounted on a motorized gantry, allowing for an additional three axis of alignment (anterior–posterior, medial–lateral and roll). Next, we utilized an optical alignment procedure consisting of a 'guide' LED light that was projected through the imaging path, reflected off the cannula cover glass and picked up by a separate CCD camera (Extended Data Fig. 2b). Using fine movement of both the microscope and the head bar, the location of the resulting intensity spot on the camera sensor could be used to ensure exact parallel alignment of the imaging plane with respect to the cover glass.

To correct for smaller shifts in the brain tissue across multiple sessions, we took a high-resolution reference z-stack at the start of the first imaging session (25 μm total, 1-μm interval; Extended Data Fig. 2c). The imaging plane on each subsequent session was then compared with this reference stack by calculating a cross-correlation in the frequency domain for each imaging stripe along all depth positions. By adjusting the scanning parameters on the remote focusing unit of the 2P-RAM microscope, we finely adjusted the tip or tilt angles of the imaging plane to achieve optimal alignment with the reference stack. We used a custom online Z-correction module (developed by Marius Pachitariu[64], now in ScanImage), to correct for z and xy drift online during the recording within each session, using a newly acquired z-stack for that specific session.

To find cells that could be consistently imaged across sessions, we first performed a post-hoc, non-rigid, image registration step using an averaged image of each imaging session (diffeomorphic demon registration; Python image registration toolkit) to remove smaller local deformations (Extended Data Fig. 2g–i). Next, we performed hierarchical clustering of detected cells across all sessions (Jaccard distance; Extended Data Fig. 3). Only putative cells that were detected in 50% of the imaging sessions were included for further consideration. We then generated a template consensus mask for each cell based on pixels that were associated with this cell on at least 50% of the sessions. These template masks were then backwards transformed to the spatial reference space of each imaging session to extract fluorescence traces using Suite2p.

### Data analysis
**Coefficient of partial determination.** To assess the unique contribution of each behavioural strategy (random licking, licking in both reward locations, lick–stop and expert) to overall animal behaviour, we used the coefficient of partial determination (CPD). In this analysis, a multivariable linear regression model was first fitted using all behavioural strategies as regressors, providing the sum of squares error (SSE) of the full model ($SSE_{fullmodel}$). Each regressor was then sequentially removed, the model refitted, and the SSE without that regressor

($SSE_{-i}$) was computed. The CPD for each regressor, denoted as $CPD_i$, was then calculated as $CPD_i = (SSE_{-i} - SSE_{fullmodel})/SSE_{-i}$, revealing the unique contribution of each behavioural strategy to the overall variance in licking behaviour.

**Place field detection.** To identify significant place cells, we utilized an approach based on Dombeck et al.[65] (but see also Grijseels et al.[66] for overall caveats with such approaches). Place fields were determined during active trials, indicated by active licking within reward zones, and at running speeds greater than 5 cm s$^{-1}$. For detecting activity changes related to position, we first calculated the calcium signal by subtracting the fluorescence of each cell mask with the activity in the surrounding neuropil using Suite2p. Next, the baseline fluorescence activity for each cell was calculated by first applying Gaussian filter (5 s) followed by calculating the rolling max of the rolling min ('maximin' filter; see Suite2p documentation). This baseline fluorescence activity ($F_0$) was used to calculate the differential fluorescence ($\Delta F/F_0$), defined as the difference between fluorescent and baseline activity divided by $F_0$. Next, we identified the significant calcium transient event in each trace as events that started when fluorescence deviated 5σ from baseline and ended when it returned to within 1σ of baseline. Here baseline σ was calculated by binning the fluorescent trace in short periods of 5 s and considering only frames with fluorescence in the lower 25th percentile.

Initially, putative place fields were identified by spatially binning the resulting $\Delta F/F_0$ activity (bin size of 5 cm) as continuous regions where all $\Delta F/F_0$ values exceeded 25% of the difference between the peak of the trial and the baseline 25th percentile $\Delta F/F_0$ values. We imposed additional criteria: the field width should be between 15 and 120 cm in virtual reality, the average $\Delta F/F_0$ inside the field should be at least four times greater than outside; and significant calcium transients should occur at least 20% of the time when the mouse was active within the field (see above). To verify that these putative place fields were not caused by spurious activity, we calculated a shuffled bootstrap distribution for each cell. Here we shuffled blocks of 10-s calcium activity with respect to the position of the mouse and repeated the same analysis procedure described above. By repeating this process 1,000 times per cell, we considered a cell to have a significant place field if putative place fields were detected in less than 5% of the shuffles.

**Population vector analysis.** For the analysis of similarity of representation between near versus far trial types, we performed population vector correlation on the fluorescence $\Delta F/F_0$ data. Each 5-cm spatial bin, we defined the population vector as the mean $\Delta F/F_0$ value for all neurons. Fluorescence data were included only when the speed of the mouse exceeded 5 cm s$^{-1}$. The cross-correlation matrix was generated by calculating the Pearson correlation coefficient between all location pairs across the two trial types.

**Spatial dispersion index.** To evaluate the extent of spatial dispersion in place tuning across single cells, such as distinguishing between tuning to single positions versus multiple positions, we took the single-cell tuning curve over track positions and normalized it so that the area under the curve is 1. The spatial dispersion index is defined as the entropy of this normalized $\Delta F/F_0$ signal by: entropy $= -\sum [p(i) \times \log_2 p(i)]$, where $p(i)$ denotes the probability associated with each position bin index.

**UMAP.** To visually interpret the dynamics of high-dimensional neural activity during learning, we utilized UMAP on our deconvolved calcium imaging data. The UMAP model was parameterized with 100 nearest neighbours, three components for a three-dimensional representation, and a minimum distance of 0.1. The 'correlation' metric was used for distance calculation. The data, a multidimensional array representing the activity of thousands of cells concatenated from several imaging sessions, were fitted into a single UMAP model. This resulted in a

three-dimensional embedding, in which each point characterized the activity of the neuron ensemble at a single imaging frame.

## Modelling

**CSCG.** In the 2ACDC task, the combination of position along the track and trial type defines a state of the world ($z$). Although this state is not directly observable to the animal, it influences the sensory observation ($x$) that the animal perceives. The sequence of states in the environment obeys the Markovian property, in which the probability distribution of the next state (that is, the next position and trial type) depends only on the current state, and not all the previous states, assuming the animal always travels at a fixed speed. When an animal learns the structure of the environment and builds a map, it tries to learn which states (position, trial type) are followed by which states, and what sensory experience they generate. This can be viewed as a Markov learning problem. A HMM consists of a transition matrix whose elements constitute $p(z_{n+1}|z_n)$ that is, the probability of going from state $z_n$ at time $n$ to $z_{n+1}$ at time $n+1$, an emission matrix whose elements constitute $p(x_n|z_n)$, that is, the probability of observing $x_n$ when the hidden state is $z_n$, and the initial probabilities of being in a particular hidden state $p(z_1)$.

The CSCG is an HMM with a structured emission matrix in which multiple hidden states, referred to as clones, deterministically map to the same observation. In other words, $p(x_n = j|z_n = i) = 0$ if $i \notin C(j)$ and $p(x_n = j|z_n = i) = 1$ if $i \in C(j)$, where $C(j)$ refers to the clones of observation $j$[21] (Extended Data Fig. 8f). The emission matrix is fixed and the CSCG learns the task structure by only modifying the transition probabilities (Extended Data Fig. 8e,f), making the learning process more efficient. The Baum–Welch expectation maximization algorithm was used to update the transition probabilities such that it maximizes the probability of observing a given sequence of sensory observations[67–69].

We trained the CSCG on sequences of discrete sensory symbols mimicking the sequence of patterns shown to the mice in the two tracks. Each 10-cm segment of the track was represented by a single sensory symbol. In addition, the teleportation region was represented by a distinct symbol repeated three times, spanning 30 cm. In the rewarded region, the mice could receive both visual input and a water reward simultaneously. However, our model could only process a single discrete stimulus at a time. Thus, we divided the rewarded region into two parts. We presented the visual cue first, mimicking the ability of the mouse to see the rewarded region ahead before reaching it. Subsequently, we presented a symbol representing the water stimulus, which was shared across the two trials. The near trial sequence, denoted as [1,1,1,1,1,1,2,2,2,2,1,1,1,4,6,1,1,1,5,5,1,1,7,0,0,0], and the far trial sequence, denoted as [1,1,1,1,1,1,3,3,3,3,1,1,1,4,4,1,1,1,5,6,1,1,7,0,0,0]′, were used. Where 1 represented the grey regions, 2 and 3 indicated the indicators for near and far tracks, respectively, 4 denoted the visual observation associated with the first reward zone, 5 represented the visual stimulus associated with the far reward zone, 6 denoted the common water reward received in both tracks, 7 represented the brick wall at the end of each trial, and 0 indicated the teleportation region (Extended Data Fig. 8c). However, the representations and learning dynamics are not sensitive to the addition of the brick wall and teleportation segments.

We initialized the model with 100 clones for each sensory observation symbol and performed 20 iterations of the expectation-maximization process at each training step with sequences from 20 randomly selected trials, comprising both near and far trial types. We extracted the transition matrix at different stages of learning and used the Viterbi training algorithm to refine the solution[21], and then plotted the transition matrix as a graph, showing only the clones that were used in the representation of the two trials (Extended Data Fig. 8a). We ran multiple simulations and compared how correlation between the two trial types changed over learning for different positions along the track (Extended Data Fig. 8b).

We also explored alternate sequences of sensory stimuli. In one variant, we provided the water symbol before the visual symbol of the reward zone (for example, [...1,1,1,6,4,1,1,1...] where 6 represented the water and 4 denoted the visual symbol). In addition, we introduced a symbol that conjunctively encoded the simultaneous water reward and visual symbol (for example, [...111,4,6,111...] in the near trial and [...111,5,8,111...] in the far trial, where 6 denoted a combined code for water and visual R1, and 8 represented a combined code for water and visual R2; Extended Data Fig. 8c). Although the final learned transition graphs matched for all the four sequence variants, the exact sequence of learning differed. Specifically, reward cue followed by a visual cue for reward zone often led to decorrelation of pre-R1 followed by pre-R2 (Extended Data Fig. 8c,d), contrary to what is often observed during learning in animals.

**Vanilla RNNs.** We implemented custom RNN models to learn the structure of the 2ACDC task. Task sequences incorporated numerical symbols with unique meanings: '1' denoted the grey region; '2' and '3' represented near and far cues, respectively; '4' and '5' indicated near and far reward cues, respectively; '6' symbolized reward; and '0' denoted teleportation. An example of a near trial followed the structure: 1,1,1,1,1,1,2,2,2,2,1,1,1,4, 6,1,1,1,5,5,1,1,0, and a far trial followed the structure: 1,1,1,1,1,1,3,3,3,3,1,1, 1,4,4,1,1,1,5,6,1,1,0. We converted these numerical symbols into one-hot encodings to represent these categories. The RNNs consisted of an input layer, a recurrent hidden layer and an output layer. Both input and output layers contained seven units, corresponding to the unique sensory cues in the task. The hidden layer size varied between 200 and 5,000 units, depending on the specific variant. We explored four activation functions for the hidden layer: exponential softmax, ReLU, polynomial softmax and sigmoid. The exponential and polynomial softmax functions implemented a soft winner-take-all mechanism, whereas ReLU and sigmoid provided more traditional activation patterns. The models were trained using backpropagation through time with the Adam optimizer. Learning rates ranged from 0.002 to 0.2, adjusted based on the activation function to ensure stable training. We used cross-entropy loss as the objective function. For each simulation, we generated sequences of 40–100 trials (random mixture of near and far trials), with half used for training and half for testing. Each trial consisted of 23 time steps corresponding to positions along the virtual track. Models were trained for 60–1,200 epochs. We ran multiple independent simulations with different random seeds to assess variability, ranging from 4 to 48 simulations depending on the specific model variant. To initialize the models, we used small random values drawn from a normal distribution for the weight matrices. The input-to-hidden and hidden-to-hidden weight matrices were initialized with a standard deviation of 0.001, whereas the hidden-to-output weight matrix used a standard deviation of 0.01–1, depending on the model variant.

**Hebbian-RNN.** Previous work[38] showed that a local Hebbian learning rule in a RNN can approximate an online version of HMM learning. We used an RNN consisting of $K = 100$ recurrently connected neurons and $n = 96$ feedforward input neurons. The feedforward input neurons carried orthogonal inputs for each of the 8 sensory stimuli, with 12 different neurons firing for each stimulus. The recurrent weights $V$ and feedforward weights $W$ were initialized from a normal distribution with 0 mean and standard deviation 2.5 and 3.5, respectively. The membrane potential of the $k$th neuron at time $t$ is given by $u_k^t = \sum_i^N w_{ki}x_i^t + \sum_j^K v_{kj}y_j^{(t-\Delta t)}$, where $w_{ki}$ is the feedforward weight from input neuron $i$ to RNN neuron $k$, $v_{kj}$ is the recurrent weight from neuron $j$ to neuron $k$, $\Delta t = 1$ ms is the update time, and $x_i^t$ and $y_j^{t-\Delta t}$ are exponentially filtered spike trains of the feedforward and recurrent neurons, respectively (exponential kernel time constant of 20 ms). The probability of neuron $k$ firing in $\Delta t$ was computed by exponentiating the membrane potential and normalizing it through a global inhibition, $f_k = \frac{e^{u_k}}{\sum_l^K e^{u_l}}$. For each neuron $k$, spikes were generated with a

probability of $f_k$ by a Poisson process, with a refractory period of 10 ms during which the neuron cannot spike again. When the postsynaptic neuron $k$ spiked, then the weights onto neuron $k$ were updated as $\Delta w_{ki}(t) = \alpha(e^{-w_{ki}}x_i(t) - 0.1)$ and $\Delta v_{kj}(t) = \alpha(e^{-v_{kj}}y_j(t) - 0.1)$, where $\alpha$ is the learning rate (0.1) and $y_j(t)$ is the exponentially filtered spike train. Both weights $V$ and $W$ were kept excitatory. We computed the correlation between the RNN representation of different positions in the near and far trial types at different stages during learning and compared it with the cross-correlation matrices for mice.

**LSTM.** We implemented LSTM networks using the same task structure and input sequences as the vanilla RNNs. The LSTM model consisted of a single LSTM layer with 500–1,200 hidden units, followed by a linear readout layer. Both input and output layers contained seven units, corresponding to the unique sensory cues in the task. The LSTM processed the input sequence and produced hidden states for each time step. These hidden states served as the primary output for analysis and were also passed through the linear readout layer to generate predictions. We explored several LSTM variants, including a standard model, one with L1 regularization on hidden states, another with dropout applied to the hidden states, and a version with a correlation penalty to encourage decorrelation between hidden states of different trial types. These models were trained using the Adam optimizer with learning rates between $3 \times 10^{-4}$ and $5 \times 10^{-4}$, using cross-entropy loss as objective function. Training proceeded for 200–300 epochs on sequences of 100 trials, with half used for training and half used for testing. We conducted multiple independent simulations with different random seeds. For all LSTM variants, we analysed the hidden state dynamics (cell states for some variants), examining their correlation structure between different trial types and the accuracy of the model in predicting reward locations.

**Transformers.** We implemented a transformer architecture based on the minGPT repository (https://github.com/karpathy/minGPT), specifically using the GPT-micro configuration. This model uses 4 layers, 4 attention heads and an embedding dimension of 256. The transformer was adapted to learn the 2ACDC task structure, using the same input encoding as the vanilla RNN and LSTM models. We generated sequences of trials with random starts, totalling 1,000–3,000 batches. Each batch consisted of ten randomly assembled trials. From these, we selected random 100-element chunks to form our input sequences. The vocabulary size was set to match our dataset, and the block size (maximum sequence length) was adjusted based on our experiments with different context lengths. To address the sequential nature of the task, we trained transformers with various context lengths ranging from 1 to 100, finding that lengths exceeding 4 were sufficient to solve the task. This threshold is specific to our task structure, allowing disambiguation between reward locations given the inter-reward grey cue length of 3. The transformer was trained using the Adam optimizer with a learning rate of $3 \times 10^{-4}$ for 600–2,000 iterations. The objective was to predict the next sensory symbol, using cross-entropy loss. During testing, we primarily used four-symbol sequences to evaluate the next-input prediction accuracy of the model. For analysis, we examined the pre-logit layer of the transformer, as it represents the final stage of feature extraction before classification, potentially capturing the most task-relevant information. Our key findings regarding the representational structure were robust across different context lengths, up to 100 symbols.

**Reporting summary**
Further information on research design is available in the Nature Portfolio Reporting Summary linked to this article.

## Data availability
Imaging data are available on Figshare[70] (https://doi.org/10.25378/janelia.27273552). We have also provided an interactive data visualization tool at http://cognitivemap.janelia.org. Source data are provided with this paper.

## Code availability
The code associated with this article is available on GitHub (https://github.com/sprustonlab/OSM_Paper_Figures).

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

**Acknowledgements** We thank H. Akhlaghpour, A. Fernandez-Ruiz, B. Hulse, A. Lee, B. Mensh, G. Michel, A. Payne, S. Romani, Y. Wang and L. Zhong for their comments on the manuscript; M. Pachitariu for assistance with the mesoscope imaging pipelines; A. Lee and J. S. Lee for their technical guidance on the CA1 window surgeries; V. Goncharov and D. Tsyboulski for mesoscope technical support; G. Michel, B. Mohar, Y. Wang, X. Zhao and other current and former members of the Spruston laboratory for their discussion, technical assistance and feedback throughout the project; S. Dilisio and S. Lindo for their assistance in animal surgeries; the Janelia Vivarium team for animal support; M. Botvinick, Z. Kurth-Nelson, D. Kumaran, K. Stachenfeld and J. Wang from DeepMind for discussions regarding artificial intelligence models; T. Behrens, F. Chollet, L. Coddington, I. Cone, J. Dudman, S. Fusi, M. Jazayeri, J. Knierim, S. Lewallen, J. Magee, B. Mensh, A. Saxe, J. Whittington and Y. Ziv for valuable discussions; the Janelia Experimental Technology team, including J. Arnold, B. Bowers, T. Goulet, D. Smith, S. Sawtelle and A. Sohn for technical assistance; J. Clements for assistance on the interactive data visualization tool; and J. Kuhl for the illustration of the virtual reality behavioural setup in Fig. 1a. This work was supported by the Howard Hughes Medical Institute.

**Author contributions** W.S., J.W. and N.S. conceptualized the core study. W.S., M.N., C.L. and J.E.F. conceptualized and performed the computational modelling. W.S., J.W., K.K., M.M. and R.G. performed the behavioural experiments and collected imaging data. W.S., J.W., M.N., K.K., A.B. and C.S. analysed the data. J.W. designed and implemented the virtual reality system and data processing pipelines. D.F. provided imaging expertise and technical support for the microscope. A.B. coded the interactive data visualization tool. N.S. directed the study throughout. W.S., J.W., M.N., J.E.F. and N.S. wrote the manuscript with input from other authors.

**Competing interests** The authors declare no competing interests.

**Additional information**
**Correspondence and requests for materials** should be addressed to Weinan Sun or Nelson Spruston.

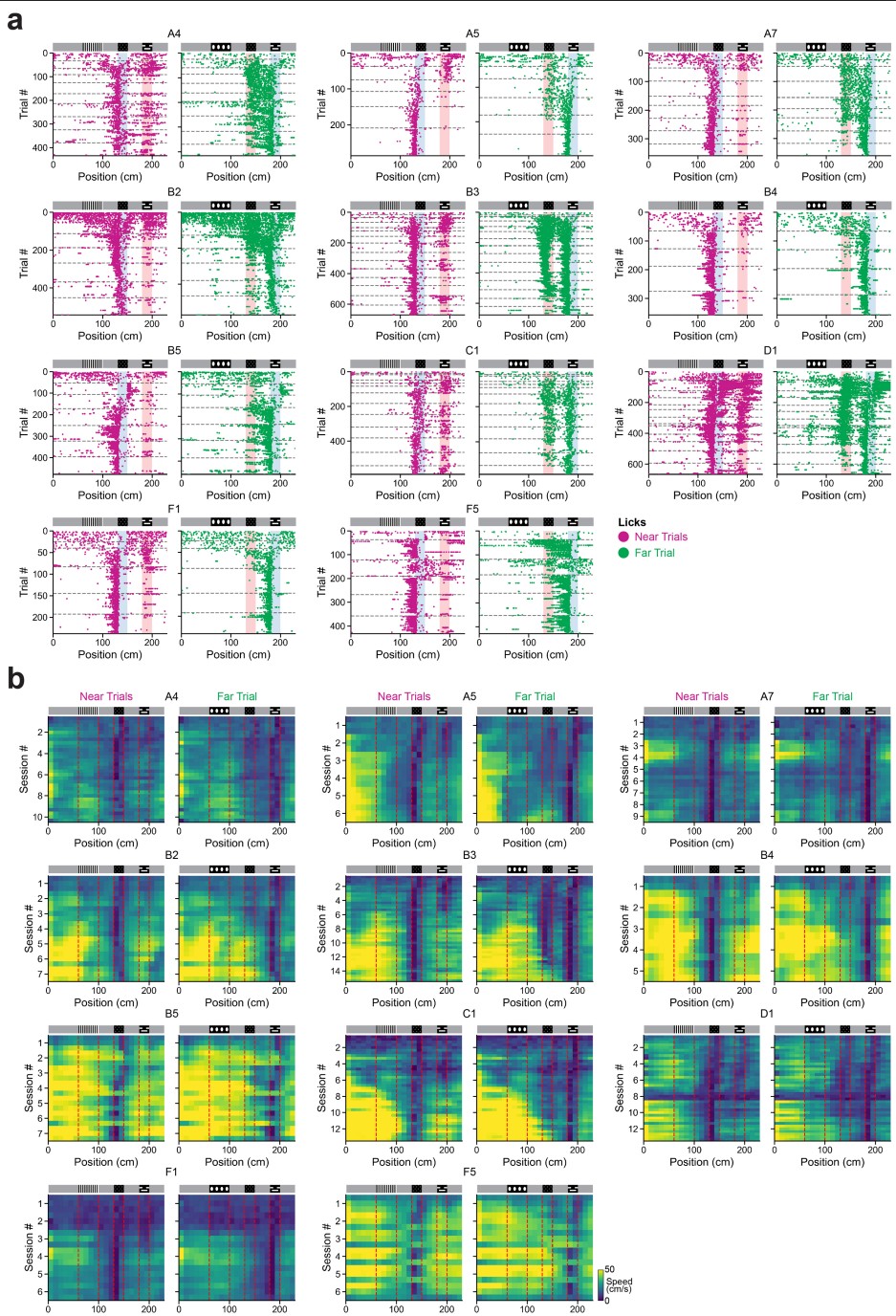

**Extended Data Fig. 1 | Changes in licking behavior and speed during learning across all animals.** (a) Each dot indicates a single lick made by an animal relative to its position on the track in either a 'near' (magenta) or 'far' trial type (green). Horizontal dashed lines indicate the end of a session. (b) Spatially binned speed profiles of all animals across training sessions during 'near' and 'far' trial types. Vertical dashed lines indicate the location of the indicator and reward cues. 'A4', 'A5'... denote animal nicknames. Blue and red shading denote correct and incorrect licking zones, respectively.

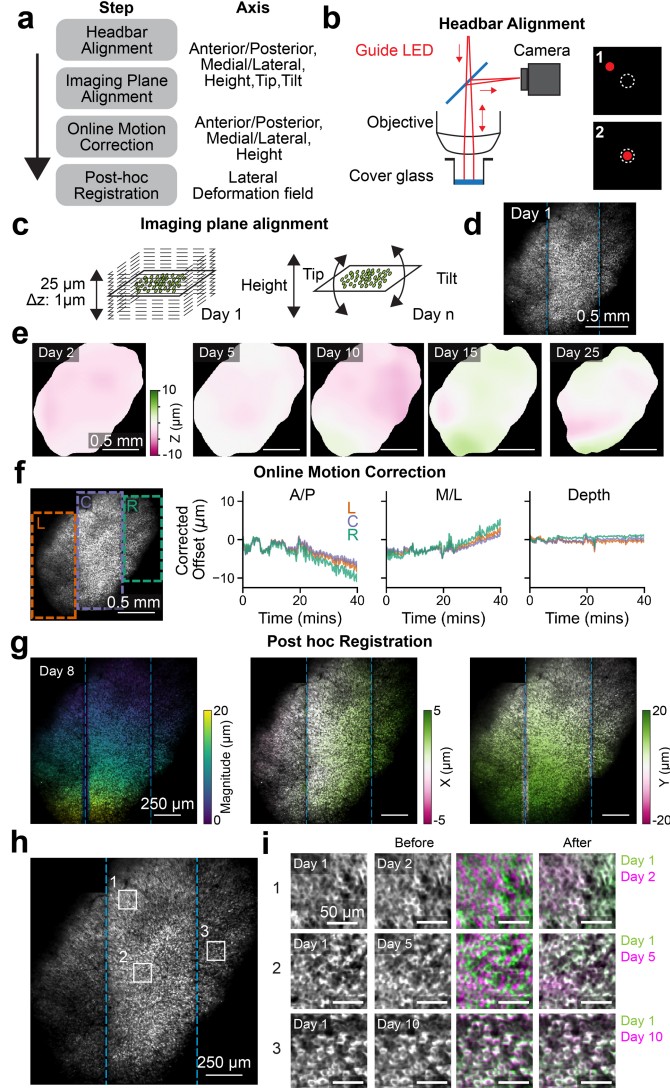

**Extended Data Fig. 2 | Overview of imaging alignment across sessions.**
(a) Left, schematic view of the alignment steps for registering the field-of-view across imaging sessions. Right, the axis that can be controlled in each alignment step. (b) Headbar alignment, focused light from a guide LED was projected through the optical path and reflected of the cover glass in the implanted canula. The position of the resulting spot on the camera sensor was used to ensure consistent alignment relative to the cover glass. (c) Image plane alignment. (left) A reference image z-stack was taken on day 1 of training. (right) The heigh, tip, and tilt of the imaging surface was adjusted on each day to achieve optimal alignment to the reference stack. (d) Example image of the field-of-view on day 1 from same animal shown in e. Dashed lines indicate the location of the imaging stripes (see Methods). (e) Heatmap of the remaining z error after alignment. (f) Online motion correction. (left) Locations of the left (orange), center (purple), and right (green) imaging stripes. (right) Online adjustment of individual imaging stripe positions during an example recording. (g) Example of post hoc registration. (left) Magnitude of elastic, non-rigid, deformation across the field-of-view. (right) Amount of deformation in x and y. (h) Location of ROIs shown in i. (i) Result of post hoc registration step. Each row shows a single ROI comparing the image on day 1 to that of day 2, 5, or 10. Third and fourth column shows the overlay of the two images with day 1 in green and the comparison day in magenta both before and after registration. Note that the outline of the cells now overlap (white pixels) indicating that the same cells can be monitored across days. Similar alignment and registration results were obtained across all 11 mice.

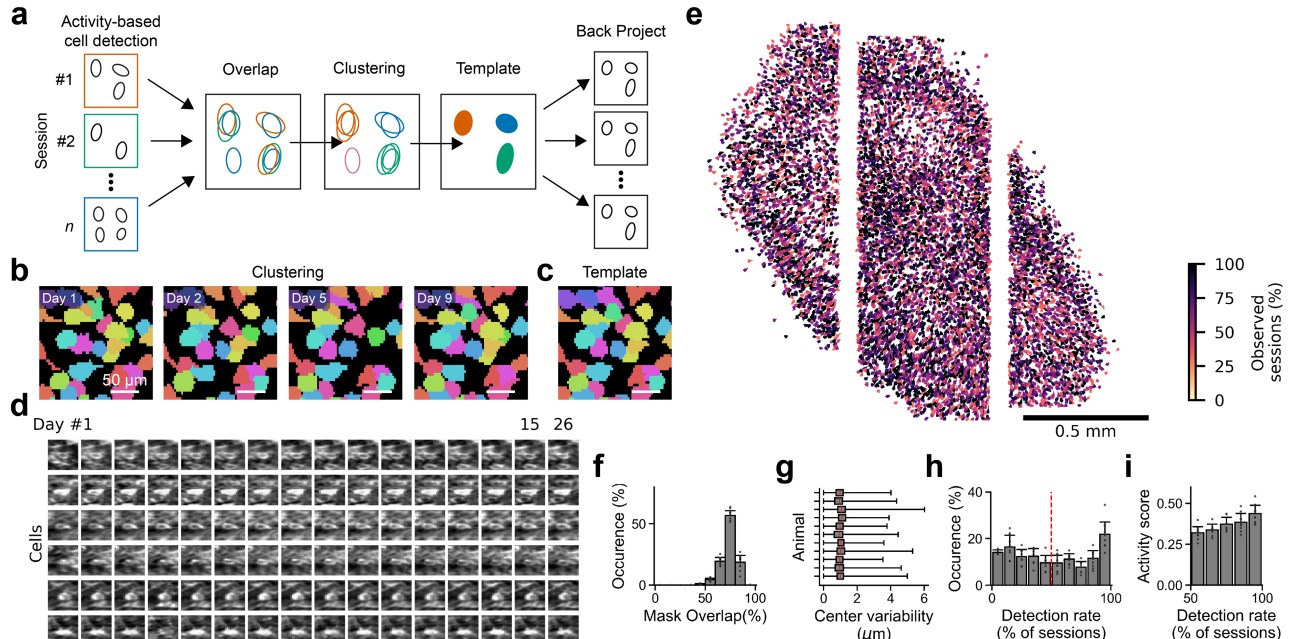

**Extended Data Fig. 3 | Identification of consistent cell masks across imaging sessions.** (a) Schematic of computational pipeline. (left) Activity-based cell mask extraction is performed for each individual session using Suite2p (see Methods). (center) The overlap between identified cell masks across all imaging sessions after registration was calculated and used to perform hierarchical clustering. The resulting clusters are used to calculate a single 'template' cell mask based on the median of present pixels across all sessions. Cell mask clusters that were not detected in the majority of sessions, or whose template mask was too small, were discarded (see Methods for additional details). (right) The template masks were projected back to the spatial reference frame of each individual imaging session and used for calcium trace extraction. (b) Example of clustered cell masks across imaging sessions. (c) Resulting cell mask templates of the same cells shown in b. (d) Same cropped region around example cells across multiple sessions. Last column is final recording session (day 26). Cells can be tracked stably across weeks. (e) Map of

all detected cells and how often they were detected across sessions. Observability was not affected by location in the field-of-view. Note that we rejected cells residing within 25 μm of the boundaries between scanning stripes. (f) Histogram of the percentage of spatial overlap of clustered cell masks with their resulting template mask averaged across animal (mean ± s.d., n = 11 animals). (g) Average variability in the center of clustered cell masks for all animals (box plots: center line, median; box limits, 25th and 75th percentiles; whiskers, range, n = 11 animals). (h) Histogram of observed detection rates of clustered cell masks as percentage of all sessions averaged across animals (mean ± s.d., n = 11 animals). Red line indicates the used inclusion threshold detection rate (50%) for all further analysis. (i) Relationship between cell detection rate and a cell activity score determined as the averaged deconvolved fluorescence signal across all sessions. Histogram values are averaged across all animals (mean ± s.d., n = 11 animals).

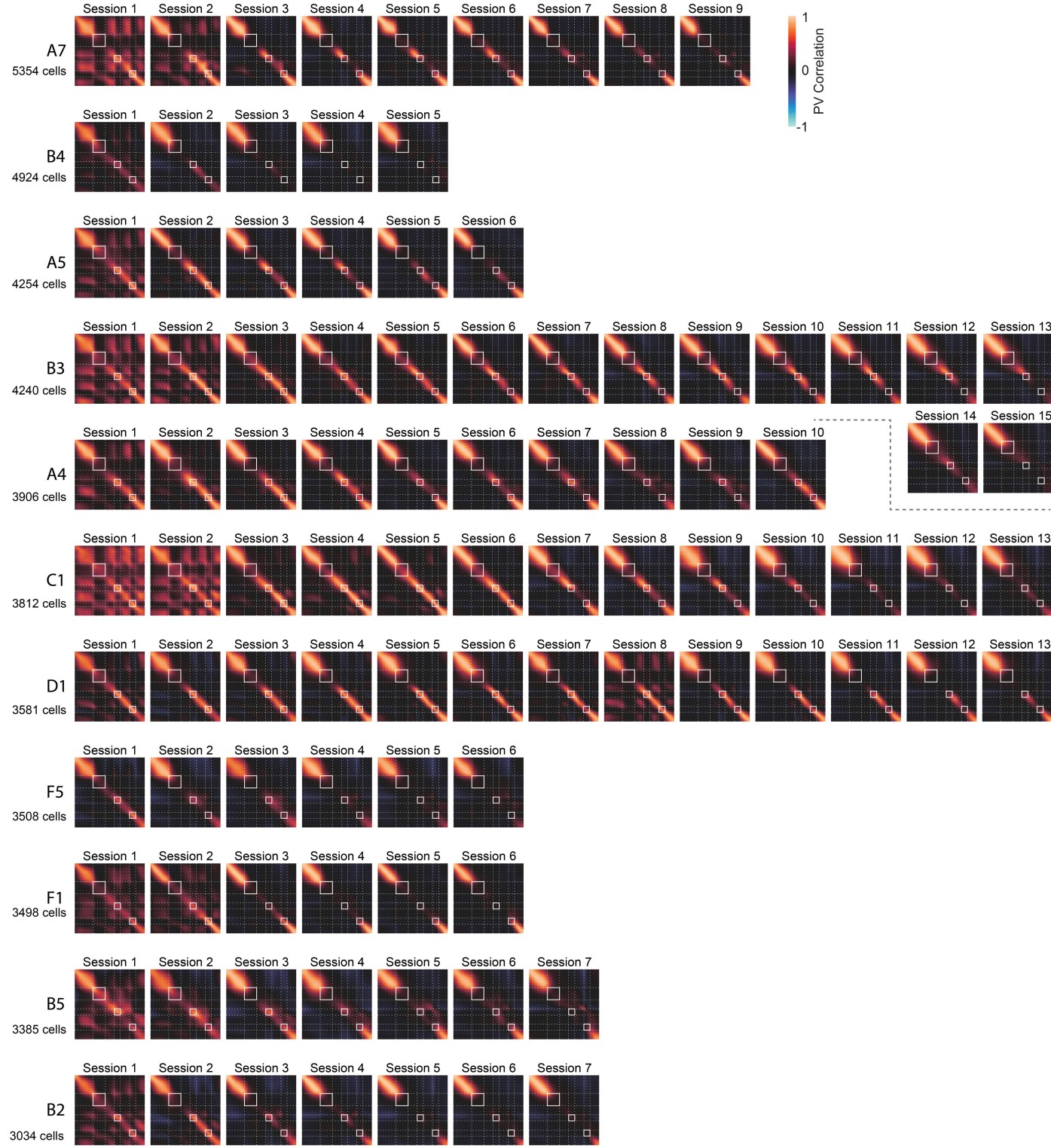

**Extended Data Fig. 4 | *Near* vs *Far* trial cross-correlation matrices for all 11 animals through all training sessions.** Animals are ordered by the number of cells registered across sessions.

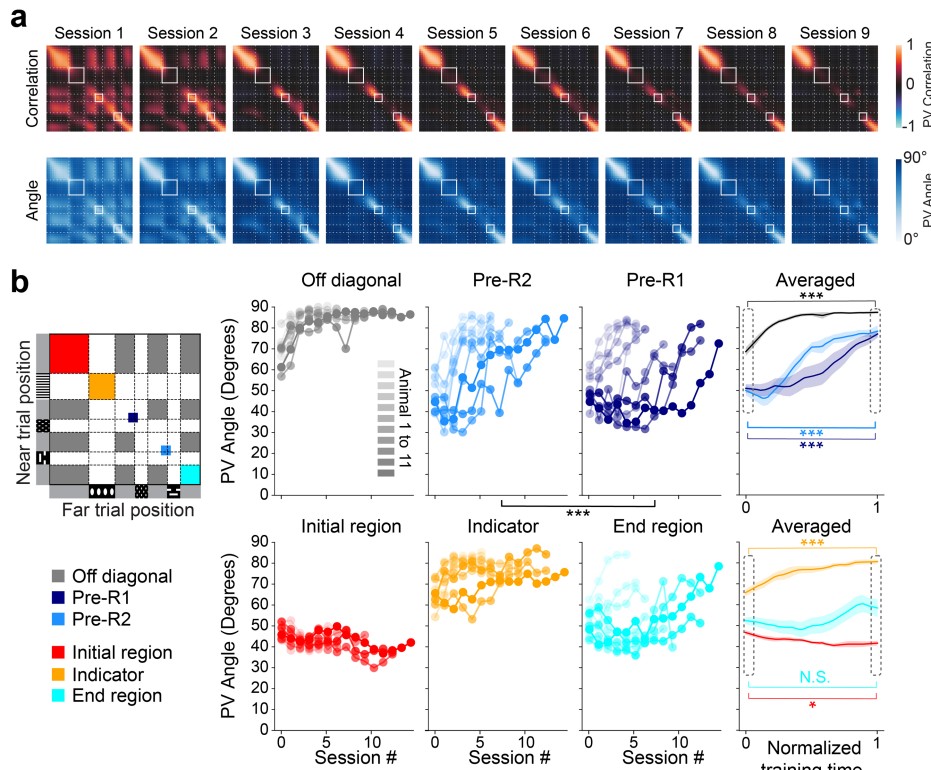

**Extended Data Fig. 5 | Changes in PV angle mirror the progressive decorrelation dynamics.** (a) Top: *Near*-versus-*Far* PV Cross-correlation matrices along all track positions for sessions 1, 2, 3, 4, 9 for an example animal. Bottom: *Near*-versus-*Far* PV angle matrices along all track positions for the same animal. (b) PV angle averaged for different regions on the angle matrix across sessions, for the off-diagonal gray region correlations (gray), pre-R2 region (light blue), and pre-R1 region (dark blue), Initial region (red), Indicator region (orange), End region (cyan) shown for each animal separately and averaged across all of them. Comparing all sessions, a significant difference was observed between the pre-R1 and pre-R2 regions (two-sided Wilcoxon signed-rank test, $P = 0.001$***, n = 11 animals). Comparisons between the first and last session revealed significant changes in Off-diagonal gray regions, pre-R2, pre-R1, Indicator (angle increasing over sessions), and Initial region (angle decreasing over sessions) with $P = 0.019$*. In contrast, changes in the End region were not significant (N.S.). These results qualitatively mirror those in Fig. 2f. Lines and shadings indicate mean ± s.e.m.

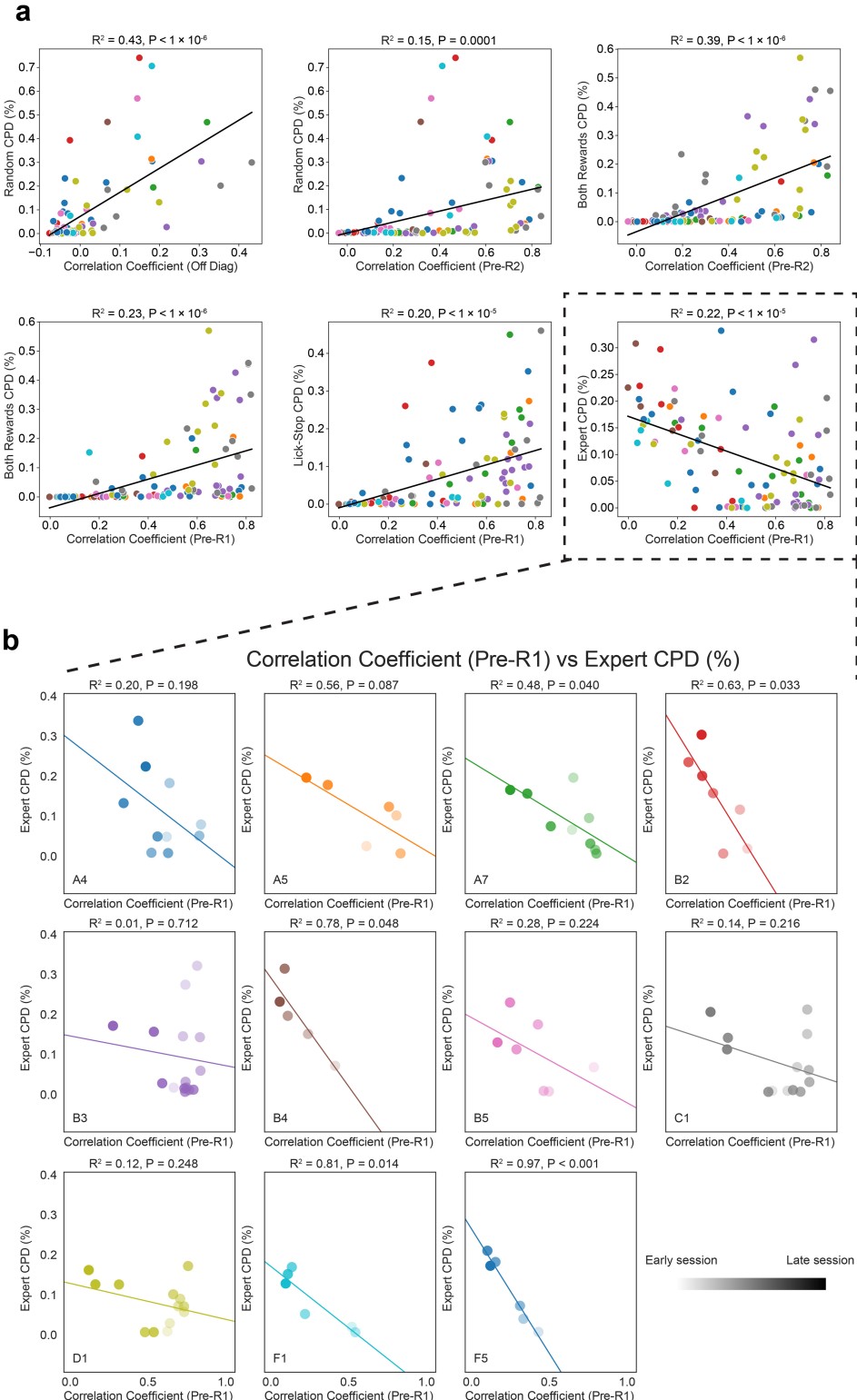

**Extended Data Fig. 6 | Licking behavior and neural activity coevolve during learning.** (a) Correlation coefficient for the off diagonal (Off Diag), pre-R1, pre-R2 regions between the *Near* and *Far* trials plotted against the CPD (%) for various basis functions for all sessions for each animal (scatter plot color coded based on individual animals). (b) Expanding the last panel in (a) into individual animals. The transparency of the filled dots indicates stage of training, with earlier sessions more transparent. Lines indicate linear regression fits, with $R^2$ and P values shown on top of each plot.

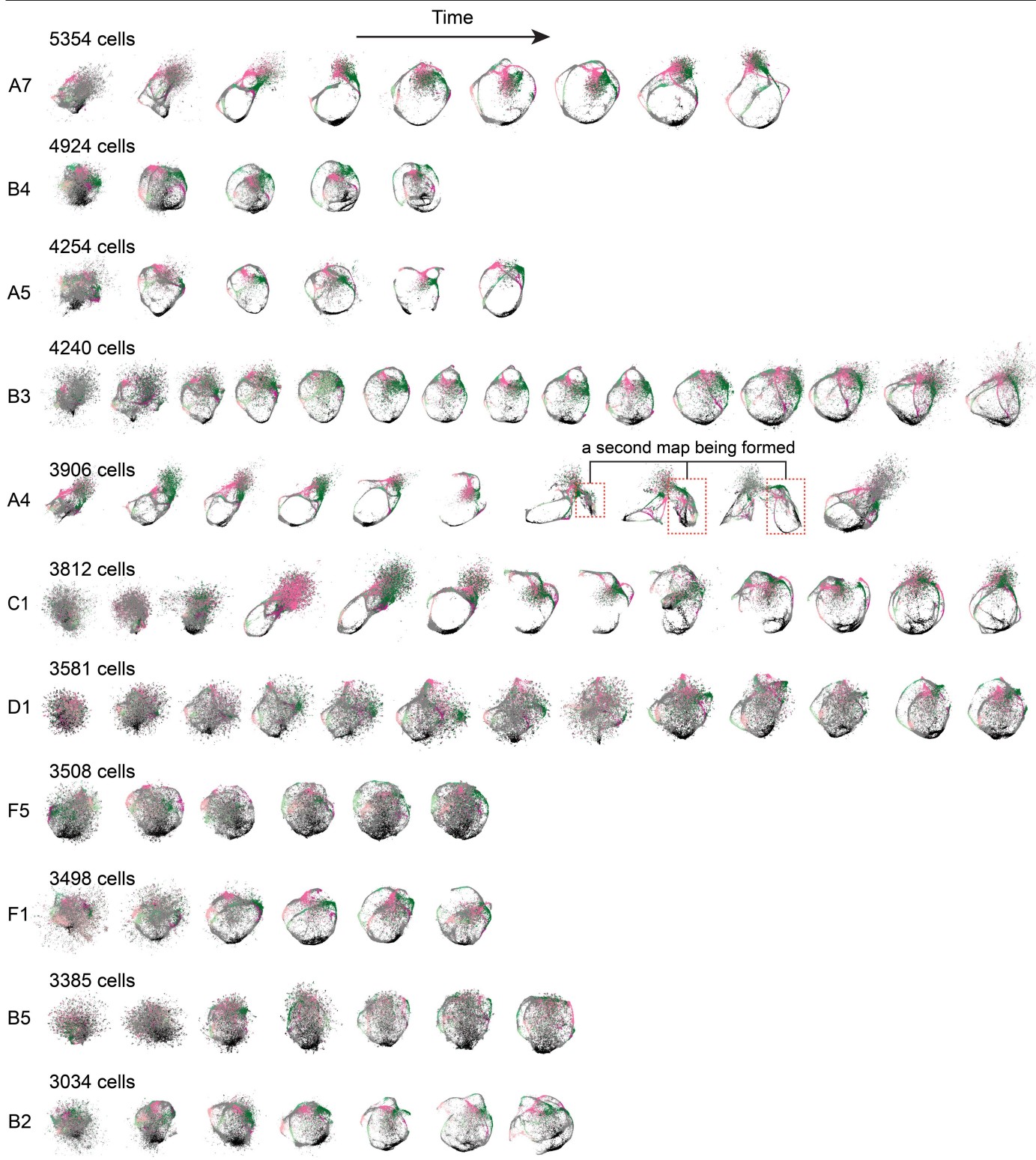

Time →

5354 cells — A7

4924 cells — B4

4254 cells — A5

4240 cells — B3

3906 cells — A4

a second map being formed

3812 cells — C1

3581 cells — D1

3508 cells — F5

3498 cells — F1

3385 cells — B5

3034 cells — B2

**Extended Data Fig. 7 | UMAP for all 11 animals through all training sessions.** Animals are ordered by the number of cells registered across sessions. Note that while UMAPs shed light on the dynamics of neural activity, our conclusions are primarily driven by the representational structure reflected by the PV angles and PV correlations. The utility of UMAP, influenced by the choice of hyperparameters and cell count, can yield a range of representations. Some may appear visually streamlined while others might seem noisy or fragmented. Even though their visual presentation may differ, these manifolds can offer potential insights into underlying neural dynamics. For example, the discovered manifolds can help reveal individual variability. In some animals, UMAP and correlation matrices both indicated lack of decorrelation at the track's end (Extended Data Fig. 4). In other cases, UMAP revealed otherwise less visible aspects, such as error trials showing single trial UMAP trajectory jumping between the embeddings of correct *Near* and *Far* trial types and a novel map appearing during learning (animal A4, this form of 'remapping' in an unchanging environment has also been observed and modeled in the entorhinal cortex[71,72]).

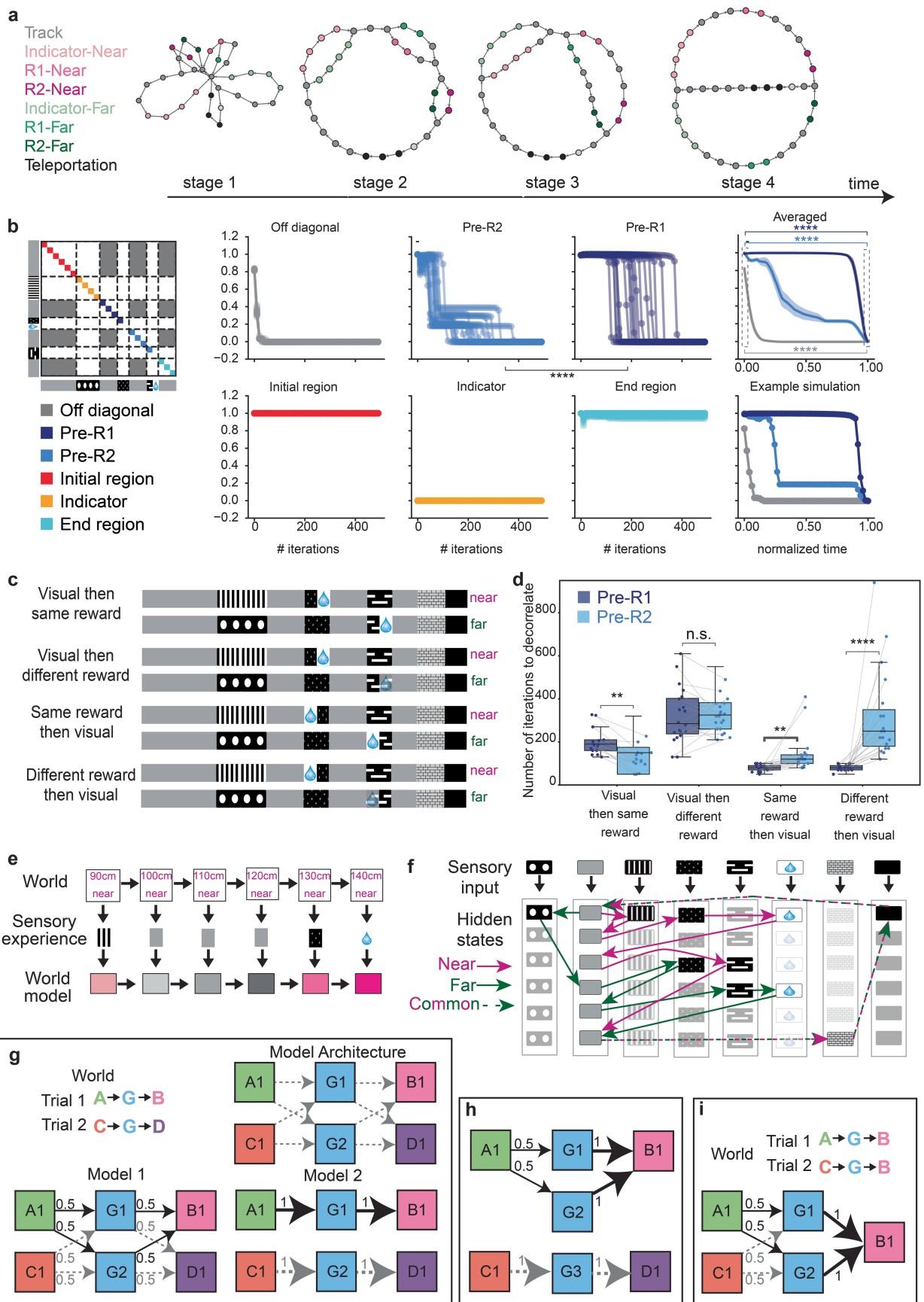

**Extended Data Fig. 8** | See next page for caption.

**Extended Data Fig. 8 | Hidden Markov learning in Clone-Structured Causal Graph recapitulates animal's learning process.** (a) The transition graph of CSCG during different learning stages recapitulates the low-dimensional neural manifolds observed in animals during learning. (b) Matrix depicting the correlation of probabilities over clones averaged for different regions: off-diagonal gray regions (gray), pre-R2 region (light blue), pre-R1 region (dark blue), Initial region (red), Indicator region (orange), End region (cyan) shown for all individual simulations that fully learned, for an example simulation, and average across all simulations (Curves represent the mean values, with shading indicating ± s.e.m). Comparing over time, a significant difference was observed between the pre-R1 and pre-R2 regions (two-sided Wilcoxon signed-rank test, $P < 0.0001$****, n = 900 datapoints compared from 18 simulations). Comparisons between beginning and end of training revealed a significant decrease in correlation for off-diagonal gray regions, pre-R2, and pre-R1 (two-sided Wilcoxon signed-rank test, $P < 0.0001$****, n = 18 simulations). (c) Schematic representation of different possible sensory symbol sequences mimicking the animal's experience, including different orders of visual and reward experiences, and a separate reward or a combined code for reward and visual. (d) Time taken for the correlation between vectors of probability over clones of pre-R1 (dark blue) and pre-R2 (light blue) between the near and far trial types to drop below 0.3. Boxplot showing the median and quartiles of the dataset, and whiskers showing 1.5 times the interquartile range. For a visual symbol followed by the same reward, the time taken to decorrelate pre-R1 significantly exceeds the time taken to decorrelate pre-R2 (n = 15 simulations, two-sided paired Student's $t$-test, $P < 0.01$**). In contrast, for other sequences, the time taken to decorrelate pre-R1 is either not significantly different from (visual then different reward, n = 20 simulations) or significantly less than the time taken to decorrelate pre-R2 ($P < 0.01$**, same reward then visual, n = 20, $P < 0.0001$****, different reward then visual, n = 19). Simulations that did not fully decorrelate both pre-R1 and pre-R2 were excluded. (e-f) Conceptual illustration of task and CSCG. (e) The world state, determined by the position and trial type, is not directly accessible to the model. Instead, the system can access sensory experiences generated based on the world state, which is used to learn a world model that accurately predicts the next sensory experience. (f) Schematic of the CSCG and the learned transition sequence. Each sensory stimulus is associated with a set of clones or hidden states. The system learns transition probabilities between these clones to generate a world model. Gray sensory stimuli are observed at distinct locations on the near and far trials, so different gray clones learn to represent these distinct locations. For less ambiguous stimuli, such as the indicator, most clones remain unused. (g-i) Toy examples illustrating orthogonalization in CSCG. (g) An example "world" comprising two sequences of observations: 'A, G, B' and 'C, G, D,' where observation G is common to both. The CSCG architecture considered includes a clone for each observation (A1, B1, etc.), except for G, which has two clones (G1 and G2). Transitions that cannot produce valid sensory sequences have been removed, leaving only the feasible transitions (gray arrows). Two model CSCGs with different transition probabilities (indicated by arrow width and numerical values) are shown. In model 1, both trials utilize both G1 and G2 clones, resulting in correlated state probabilities for G across the two trials. When the first observation is A, the sequence 'A, G, B' can be generated through two latent state sequences: A1 → G1 → B1 and A1 → G2 → B1 (black arrows), each with a probability of 0.25, leading to an overall probability of 0.5. This lower probability arises because this model could also produce unobserved sequences like 'A, G, D'. In model 2, when the first observation is A, the sequence 'A, G, B' is generated by a single latent sequence: A1 → G1 → B1 with a probability of 1. The alternative sequence 'A, G, D' has a probability of 0. This transition matrix maximizes the likelihood of observed sequences in the toy world by utilizing G1 and G2 clones separately for each trial, thereby orthogonalizing the representation of G across the two trials. (h) Illustration of an HMM with a different architecture with 3 latent clones for observation 'G'. The transition matrix depicted uses multiple clones 'G1' and 'G2' for the 1st trial, yet it maximizes the observation sequence by utilizing distinct clones across the two trials ('G1, G2' vs 'G3'). This suggests that representations must be orthogonal, but not necessarily highly sparse. (i) A different example "world" consisting of two sequences of observations: 'A, G, B' and 'C, G, B,' where the observation G appears after distinct cues ('A' vs. 'C') but is followed by the same cue ('B'). Illustration of a particular transition matrix, where both trials utilize G1 and G2 clones. If the first observation is A, the sequence 'A, G, B' can be generated through two latent state sequences: A1 → G1 → B1 and A1 → G2 → B1 (black arrows), each with a probability of 0.5, which results in a combined probability of 1, despite correlated representations of G across the two trials. Since G is followed by the same observation ('B'), it is possible to maximize the probability of observation sequence without needing to decorrelate the representation of G. This helps explain why the end of the track remains correlated across near and far trials in many animals.

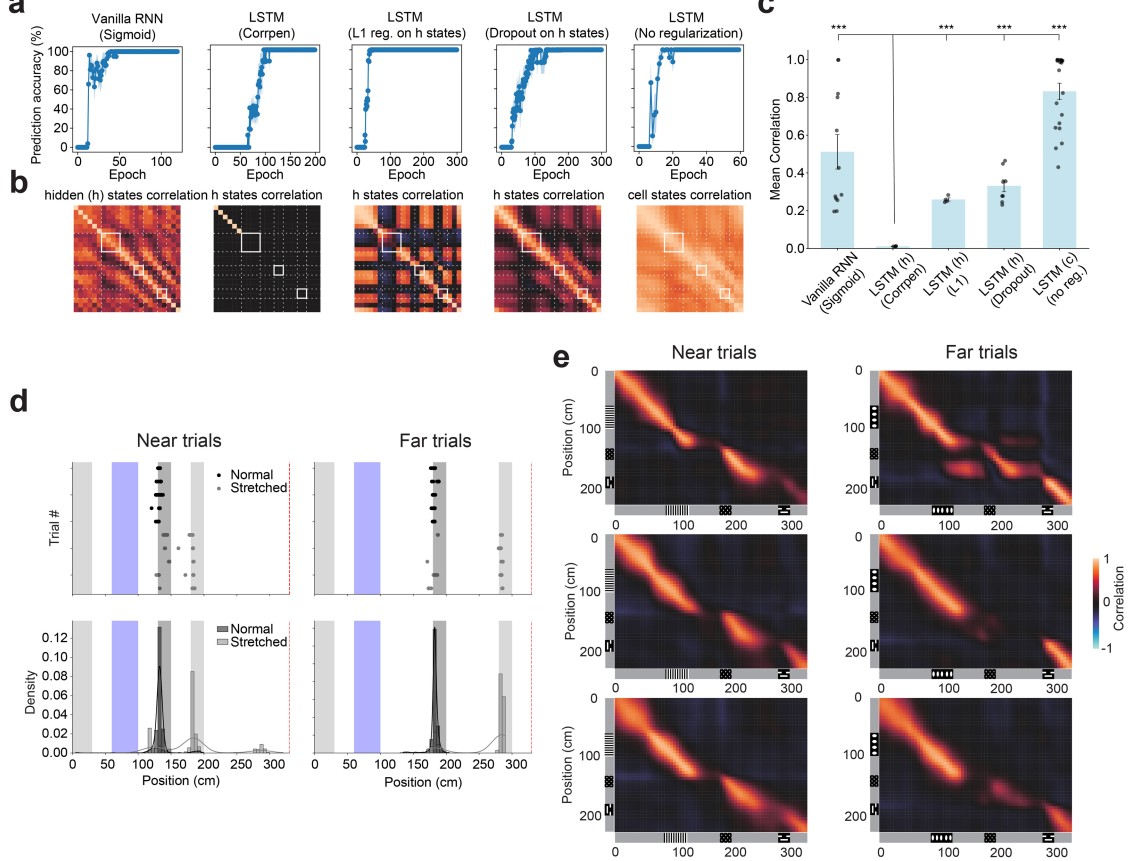

**Extended Data Fig. 9 | Model Comparisons and analysis of behavioral and neural activity during stretched trials.** Reward symbol prediction accuracy (a) and final correlation matrices (b) for various models. (c) Quantification of the mean final correlation matrix. Data are presented as mean ± s.e.m. Regularization strength was incremented progressively; the final level was selected when subsequent increase began to degrade test performance (Regularization strength: Correlation penalization 'Corrpen': 0.1; L1: 2; Dropout: 0.5). Correlation penalization involved storing hidden state activations for both a Near and a Far trial. The sum of all entries within the cross-correlation matrix between the two trial types was then added to the training loss. Bar graph showing mean ± s.e.m, *, **, *** indicate P < 0.05, 0.01, 0.001, respectively (two-tailed, unpaired Student's *t*-test, number of independent simulations: n = 12 for Vanilla RNN (sigmoid); n = 4 for LSTM (corrpen); n = 8 for LSTM(dropout); n = 20 for LSTM (no regularization)). (d) Example licking patterns (top row) and the licking position distribution over a single session (bottom row) in both near and far trials for normal (black) and stretched (gray) trials. (e) PV Correlation between the average neural population activity in normal and in stretched trials for both near (left column) and far (right column) trials. Each row corresponds to a single animal.

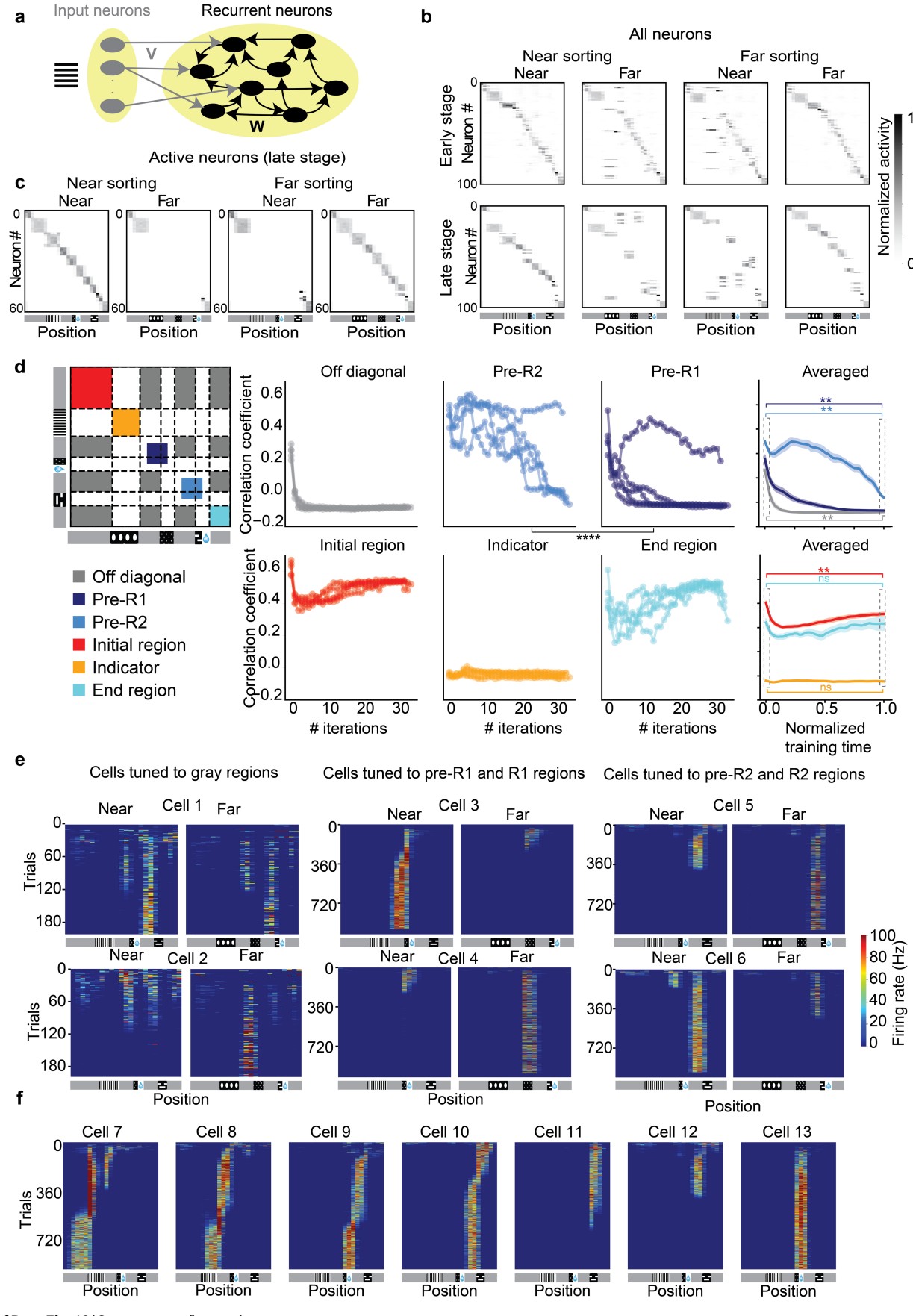

**Extended Data Fig. 10** | See next page for caption.

**Extended Data Fig. 10 | Hebbian-RNN recapitulates learned representations of animals at the population and single-cell level, though the precise learning trajectory differs from animals.** (a) Schematic representation of a recurrent neural network (RNN) used to model the hippocampus. (b) Trial-averaged neural activity plotted against track position for both near and far trial types, at early and late stages of learning. Left: Cells ordered by their activity in the near trial type. Right: Cells ordered by their activity in the far trial type. Initially, the same cells encode both trial types (except the indicator region), but as learning progresses, cells coding for regions from the indicator to R2 become trial type specific. (c) Same as (b), but only showing the active firing cells for the expert stage of near (left) and expert stage of far (right). (d) Near vs far PV Matrix depicting the correlation of probabilities over clones averaged for different regions: off-diagonal gray regions (gray), pre-R2 region (light blue), pre-R1 region (dark blue), Initial region (red), Indicator region (orange), End region (cyan) shown for all individual simulations and average across all simulations (Curves represent the mean values, with shading indicating ± s.e.m). Comparing over time, a significant difference was observed between the pre-R1 and pre-R2 regions (two-sided Wilcoxon signed-rank test, P < 0.0001****, n = 400 datapoints from 8 simulations). Here, the pre-R1 region (navy blue) decorrelates before the pre-R2 region (sky blue), an order different from that observed in most animals. Comparisons between beginning and end of training revealed no significant difference in correlation for indicator and end region but a decrease for the initial region (two-sided Wilcoxon signed-rank test, P < 0.01**, n = 8 simulations). However, the change in correlation appears to be non-monotonic with an initial decrease and subsequent increase for Initial and End regions of the track. (e) Dynamics of positional tuning for RNN cells replicate aspects of the single-cell dynamics observed in animals. Left: Example cells involved in the transition from stage 1 to stage 2, where neurons tuned to multiple gray regions become selective to one. Middle: Example cells tuned to pre-R1 and R1 regions for both trial types become selective to one trial type. Right: Example cells tuned to pre-R2 and R2 regions for both trial types become selective to one trial type. (f) Example cells exhibiting selective firing at various locations along the track in the near trial type. This includes a backward shift in cells 7 to 10, loss of selectivity in cells 11 and 12, and a stable field in cell 13.

# Reporting Summary

## Statistics

For all statistical analyses, confirm that the following items are present in the figure legend, table legend, main text, or Methods section.

| n/a | Confirmed | |
|---|---|---|
| ☐ | ☒ | The exact sample size (*n*) for each experimental group/condition, given as a discrete number and unit of measurement |
| ☐ | ☒ | A statement on whether measurements were taken from distinct samples or whether the same sample was measured repeatedly |
| ☐ | ☒ | The statistical test(s) used AND whether they are one- or two-sided<br>*Only common tests should be described solely by name; describe more complex techniques in the Methods section.* |
| ☒ | ☐ | A description of all covariates tested |
| ☒ | ☐ | A description of any assumptions or corrections, such as tests of normality and adjustment for multiple comparisons |
| ☐ | ☒ | A full description of the statistical parameters including central tendency (e.g. means) or other basic estimates (e.g. regression coefficient) AND variation (e.g. standard deviation) or associated estimates of uncertainty (e.g. confidence intervals) |
| ☐ | ☒ | For null hypothesis testing, the test statistic (e.g. *F*, *t*, *r*) with confidence intervals, effect sizes, degrees of freedom and *P* value noted<br>*Give P values as exact values whenever suitable.* |
| ☒ | ☐ | For Bayesian analysis, information on the choice of priors and Markov chain Monte Carlo settings |
| ☒ | ☐ | For hierarchical and complex designs, identification of the appropriate level for tests and full reporting of outcomes |
| ☐ | ☒ | Estimates of effect sizes (e.g. Cohen's *d*, Pearson's *r*), indicating how they were calculated |

*Our web collection on statistics for biologists contains articles on many of the points above.*

## Software and code

Policy information about availability of computer code

| Data collection | ScanImage® 2021.0.0, Matlab R2021a |
|---|---|
| Data analysis | Python 3.8, Suite2p 0.10.3, Matlab R2021a, and custom code for VR logging: https://github.com/winnubstj/Gimbl and generating the figures: https://github.com/sprustonlab/OSM_Paper_Figures |

For manuscripts utilizing custom algorithms or software that are central to the research but not yet described in published literature, software must be made available to editors and reviewers. We strongly encourage code deposition in a community repository (e.g. GitHub). See the Nature Portfolio guidelines for submitting code & software for further information.

## Data

Policy information about availability of data

All manuscripts must include a data availability statement. This statement should provide the following information, where applicable:
- Accession codes, unique identifiers, or web links for publicly available datasets
- A description of any restrictions on data availability
- For clinical datasets or third party data, please ensure that the statement adheres to our policy

Imaging data are available at Figshare: https://doi.org/10.25378/janelia.2727355267. We also provide an interactive data visualization tool at http://cognitivemap.janelia.org.

# Research involving human participants, their data, or biological material

Policy information about studies with human participants or human data. See also policy information about sex, gender (identity/presentation), and sexual orientation and race, ethnicity and racism.

| | |
|---|---|
| Reporting on sex and gender | N/A |
| Reporting on race, ethnicity, or other socially relevant groupings | N/A |
| Population characteristics | N/A |
| Recruitment | N/A |
| Ethics oversight | N/A |

Note that full information on the approval of the study protocol must also be provided in the manuscript.

# Field-specific reporting

Please select the one below that is the best fit for your research. If you are not sure, read the appropriate sections before making your selection.

☒ Life sciences  ☐ Behavioural & social sciences  ☐ Ecological, evolutionary & environmental sciences

For a reference copy of the document with all sections, see nature.com/documents/nr-reporting-summary-flat.pdf

# Life sciences study design

All studies must disclose on these points even when the disclosure is negative.

| | |
|---|---|
| Sample size | Sample sizes were determined based on previous studies investigating hippocampal neural activity during learning in head-fixed mice. No explicit power analysis was performed, but the number of animals and recorded neurons was comparable to or exceeded those used in similar published work in the field. |
| Data exclusions | 4 mice that did not reach expert performance within 15 days of task training were excluded from analysis. Imaging for 5 mice with damaged or degraded imaging window were terminated and the data were excluded from the study. |
| Replication | The experiments were independently replicated across four different groups mice by five different team members (including 3 postdocs, two technicians). Each experimenter independently trained and tested their own mouse/mice on the task. While individual animals showed some behavioral variations, the key qualitative findings were consistent across all cohorts and experimenters. The data analysis procedures and main findings were independently validated by at least five co-authors, each replicating the key results. |
| Randomization | Our study was observational and focused on tracking the progression of neural activity over time in a group of animals. Since there were no interventions or treatment groups to assign, and all subjects underwent the same imaging process, randomization was not necessary or applicable for our research objectives. |
| Blinding | Blinding was not employed in this observational study as it involved direct imaging of neural activity, where observer bias is unlikely and blinding is not typically applicable. |

# Reporting for specific materials, systems and methods

We require information from authors about some types of materials, experimental systems and methods used in many studies. Here, indicate whether each material, system or method listed is relevant to your study. If you are not sure if a list item applies to your research, read the appropriate section before selecting a response.

## Materials & experimental systems

| n/a | Involved in the study |
|---|---|
| ☒ | Antibodies |
| ☒ | Eukaryotic cell lines |
| ☒ | Palaeontology and archaeology |
| ☐ | ☒ Animals and other organisms |
| ☒ | Clinical data |
| ☒ | Dual use research of concern |
| ☒ | Plants |

## Methods

| n/a | Involved in the study |
|---|---|
| ☒ | ChIP-seq |
| ☒ | Flow cytometry |
| ☒ | MRI-based neuroimaging |

# Animals and other research organisms

| | |
|---|---|
| Laboratory animals | Thy1-GCaMP6f transgenic mice (JAX line GP5.17) were used in this study, aged 3-6 months at the time of surgery (3-8 months at the beginning of imaging studies). Mice were housed in a temperature and humidity-controlled vivarium maintained at 21 ± 1°C and 50 ± 10% relative humidity, with a reversed 12-hour light-dark cycle. |
| Wild animals | This study did not involve wild animals. |
| Reporting on sex | Mice of both sexes were included in the study. |
| Field-collected samples | No field collected samples were used in the study. |
| Ethics oversight | Janelia Research Campus Institutional Animal Care and Use Committee. |

Note that full information on the approval of the study protocol must also be provided in the manuscript.

# Plants

| | |
|---|---|
| Seed stocks | *Report on the source of all seed stocks or other plant material used. If applicable, state the seed stock centre and catalogue number. If plant specimens were collected from the field, describe the collection location, date and sampling procedures.* |
| Novel plant genotypes | *Describe the methods by which all novel plant genotypes were produced. This includes those generated by transgenic approaches, gene editing, chemical/radiation-based mutagenesis and hybridization. For transgenic lines, describe the transformation method, the number of independent lines analyzed and the generation upon which experiments were performed. For gene-edited lines, describe the editor used, the endogenous sequence targeted for editing, the targeting guide RNA sequence (if applicable) and how the editor was applied.* |
| Authentication | *Describe any authentication procedures for each seed stock used or novel genotype generated. Describe any experiments used to assess the effect of a mutation and, where applicable, how potential secondary effects (e.g. second site T-DNA insertions, mosiacism, off-target gene editing) were examined.* |

