## [Peer Review File · Nature]

Learning produces a hippocampal cognitive map in the form of an orthogonalized state machine

Corresponding Author: Dr Nelson Spruston

Version 1:

Reviewer comments:

Referee #1

(Remarks to the Author)

The manuscript "Learning produces a hippocampal cognitive map in the form of an orthogonalized state machine" reports a longitudinal observation of thousands of hippocampal cells as animals learn to operate in a two virtual environments. The number of cells observed, while still only about 1-2% of the total number of neurons in the area, is staggeringly high, especially for a longitudinal study. By diving into this large data, the authors found that the hippocampus gradually represents distinct states with different response patterns. This decorrelation appears to reflect the learning behavior. The authors then asked whether such a feature is unusual or not, by training different machine learning and neuroscience models and testing whether the hidden states follow this tendency to de-correlate representation through learning. They found that transformers and LSTMs trained with backprop did not while a particular type of hidden Markov model and a Hebbian spiking neural network did.

The study addresses a crucial question of interest to most neuroscientists: how does learning proceed in a large population? I think this paper contributes two rather important observations. One is that the learning of the cognitive maps is associated with a stage-like decorrelation of the representation, as both the behavior and the representations appear to switch between relatively stable states. The other is that the gradual process of decorrelation is uncommon in sequence learning algorithms, but natural in cognitive models and spiking neural networks. These findings are consistent with a lot of what we know of the hippocampus and machine learning models. I don't know whether the findings will be considered as obvious or not to most readers, but I think it is steering the field in the right direction.

The manuscript is particularly well written. There is an attention to detail and a clarity of the exposition that is commendable. The unusually long discussion makes the findings more salient by making well-read comparisons with various parts of the literature. The data appears to me of excellent quality. The analysis and modeling would, however, require further attention (see below).

Some aspects require further attention:

1. What is being compared? The article is framed around the question of learning. Comparing the features found in the data with computational models could shed light on the brain's learning algorithm. The failure of LSTMs and transformers is flagged with the 'gradient based' aspect of their learning algorithms. However, across the models considered in the article, many conceptually relevant aspects change: architecture, model goal (e.g. a model of cognition or a model for sequence learning) and learning algorithms (e.g. STDP, backprop, EM). I find the paper is vague about what is the main conclusion of this comparison. Is it simply to show that decorrelation is not trivial? Or is it to conclude that we should take one algorithm over another for the hippocampus and why?

1.2 LSTMs and Transformers are tagged as 'gradient based' in opposition to the other two models, implying that this aspect of the learning algorithm is what distinguishes these models. However, EM is also a gradient-based algorithm as it is a method to maximize the likelihood by an approximation that switches between two categories of parameters, iterating between two gradient descent steps. EM may not be the backpropagation of error algorithm, but maybe backprop is not the right algorithm for cloned HMMs. In any case, the emphasis on gradient-based as a distinguishing factor seems ill-placed.

1.3 In the context of what is being compared, the STDP RNN choice is also confusing. One would assume that the authors wanted to include a bottom-up model of the hippocampus. But the authors will agree that a RNN is the wrong model for CA1. Perhaps the authors wanted to include the RNN as a model of CA3, but it should really be made explicit (by adding a

readout layer and calculating the PV-correlation on the readout layer). Note that Diehl and Cook studied a similar model as Kappel and Maass but without recurrence, which would be appropriate here.

1.4 The comparison sits on the PV-correlation across near and far, both in algorithms and data. While the data samples from CA1 only (and not from CA3, DG or dorsal visual cortex), where are the models sampled from? I could not find any indication of this in the methods. It would seem that sampling from the readout layer or the hidden states of a transformer would yield widely different results. Transformers and LSTMs not being models of the hippocampus, there is not logical place to constrain the comparison, yet this can have implications on the resulting correlation structure.

2. Stage-like? There are many mentions of stage-like learning, both in terms of representation and behavior. However, I could not find any quantification of this. This is not a central claim of the paper (although it is an interesting observation). The paper nonetheless contains claims that should be supported by evidence or removed. Behaviour seems stage like, but it is not quantified. Representations gradually decorrelate but stage-like would imply a significant difference between the progression of Pre-R1 and Pre-R2 in Fig 2f, which seems to be the case but is not tested. Ultimately, whether the stages in representations correspond to stages in behavior is not tested either.

3. Transformers. While LSTMs is a logical choice for sequence learning, the choice of transformers is less clear. I also found that the description in the methods was insufficient to understand how the transformers were implemented. In particular, I am puzzled by how the authors forced the algorithm to receive one input location at a time. Transformers normally spatialize sequences and need to wait for the whole sequence to provide an output, a feature that would appear to be antagonistic with the requirement of the task.

4. Splitter cells. Is the decorrelation observed 100% synonymous with the concept of splitter cells? If you'd remove all the splitter cells, what correlation patterns would remain? This is an important question to assess the novelty of the experimental findings.

5. Quantification. There seem to be no quantification of the difference between mouse and the models. The heat maps are presented as qualitative evidence.

Minor

6. The supplementary figure on STDP+RNN is not consistent as different orderings of activity shows dramatically different levels of activity. (E.g. the near sorting in the near trial shows a diagonal, but the near trial in the far sorting shows mostly white; we would expect scattered activity, unless there is just very little activity on that time scale). It is not clear to me that this network requires the recurrent excitatory connections to capture the decorrelation. Diehl and Cook showed surprising results based on Kappel et al. but in the absence of recurrent connection (and not for HMMs).

7. Germane to the idea of spotting learning rules by looking at the evolution in neuronal activity are: Nayebi et al. Neurips "Identifying learning rules from neural network observables" (2020), Wang et al., BioRxiv "inferring plasticity rules from single-neurons" (2023); the present paper can be seen as a feature-matching approach instead of ML classification.

8. Other concepts in machine learning may relate to this question. I am thinking of contrastive losses (also related to neuroscience now, see Halgaval and Zenke Nature Neurosci (2023), which are more naturally including the concept of orthogonalization. Another concept in machine learning is object- and agent-centric representations. These ideas may lie beyond the scope of the present work, but it is likely that these techniques capture decorrelation in a natural way while using backprop for training.

9. A small thing to check, but it seems that the LSTMs would have a different match if the labels for a given position were not repeated multiple time ([1,2...] instead of [1,1,1,1,2,2,2,2...]).

Referee #2

(Remarks to the Author)

In this paper, Sun and colleagues propose that area CA1 of the hippocampus supports learning of an "orthogonalized state machine", where representations of task states become progressively decorrelaed. They present evidence for this proposal using calcium imaging data of many CA1 neurons in mice. Their modeling shows that only an HMM-like model acquires the decorrelated representations characteristic of the neural data.

Overall, I thought this was an interesting and well-written paper, though I'm not sure it represents a huge leap beyond previous data. The idea that the hippocampus does some form of structure learning, and is capable of pattern separation, is well-established (as the authors note). What's new here is the large-scale and longitudinal aspect of the data, which allowed the authors to reveal the gradual emergence of structure. I found that aspect novel and exciting.

I felt that the paper was limited in a few ways. First, the experimental design was rather unimaginative relative to the complexity of the models under consideration. Second, the approach to modeling was not as comprehensive as one might like; there was no evaluation (as far as I could tell of) parameter sensitivity, no model fitting, no formal model comparison. The interpretation rests almost entirely on qualitative visual inspection. Third, and more importantly, there was no attempt to elucidate the underlying principles explaining why some models match better than others. The considered models are very different from each other; to explore the necessary and sufficient conditions for matching neural data, the authors would need to develop systematic variants of the winning model. Finally, the authors do not evaluate any of the models on the test conditions (new cues and stretched trials). I suspect this evaluation would be highly informative about the adequacy of the models.

Minor comments:

In the figure captions, please state what the error bars show.

p. 12: "representational structure of cognitive map" -> "representational structure of the cognitive map"

p. 30: "used Viterbi training algorithm" -> "used the Viterbi training algorithm"

Referee #3

(Remarks to the Author)

I thoroughly enjoyed reading this paper. The findings are important not only for the neuroscience community, but also for the community of AI researchers that are looking to the neuroscience field for inspiration. The paper has major strengths in several aspects, and is a well done study that I'd strongly recommend for publication in Nature.

This is one of the first experiments that brings out the role of perceptual ambiguity, and how the different stages of learning in the hippocampus overcome this perceptual ambiguity in an experimental setting in a proper way. Animals get only sensations from the world — they do not get GPS coordinates. How an animal recovers representations like "locations" from this egocentric sensory experience is one of the main mysteries behind hippocampus representations. The experimental setting in this paper goes at the heart of this question, more thoroughly than previous studies. Earlier studies were limited by the number of neurons recorded or by the ability to track them across long durations. This paper overcame those limitations to perform a set of well designed experiments that ask and answer important questions in the field while also opening up new and interesting directions for future investigations.

Moreover, their computational studies are well done and powerful. One drawback of many experimental neuroscience papers is that they experimental neuroscientists often hand-craft a model based on a theory they arrived from the paper. Refreshingly, this paper is doing the opposite. They have the observations, but they don't try to fit a new theory, or add bells and whistles to an existing model. Instead, they rigorously test how the model compares against well known paradigms in machine learning, one of which also include a model called CSCG, an ML model that is also gaining attention as model of the hippocampus. The studies are well done, with proper controls, regularizations, and observations. and the conclusions are important for multiple communities. One of the striking aspects of the paper to me was that the researchers paid careful attention to their experiments to pick up learning dynamics differences that are in some ways central to some of the debates on internal representations, world models, uncertainty, and planning. I mention some of the details of this in the detailed review below.

One model that is missing from the comparisons is the Tolman-Eichenbaum Machine (TEM), and readers might be curious why. It would be good to address this in the manuscript itself. I do not think a new experiment is necessary. It might be worth mentioning that TEM doesn't deal with perceptual ambiguity from exposure to one environment. It needs exposure to multiple environments sharing the same structure to overcome the aliasing of the kind in this experimental setting. Moreover, TEM also requires an allocentric action map, which is not the case with the teleportation setting in this experiment — this could be another reason why TEM might not be applicable. Similarly, Successor Representations (SR) is another model that is popular among hippocampus researchers. I think it is worth pointing out to the readers that SR doesn't deal with perceptual ambiguity and is only applicable in settings where locations are assumed to be directly sensed. Again, no new experiment is necessary because the SR model is simply not applicable in this setting.

More detailed comments from the read through.

Line 107: "short term memory of the indicator cue after it disappears "

Yes, this is an important point. But it might also be worth point out at this stage that, as it turns out, this short term memory is carried forward by creating latent states.

Figures 2c and 2d nicely indicate the sensory ambiguity that is reflected in the early learning stage, and then how they decorrelate after learning.

It is impressive to see how the behavior, the hypothesized state diagram for the behavior, and the neural states correspond well at different stages of learning. It is also striking to see that it is possible to do this kinds of experiment that can really help us select between different theories. Similarly, the UMAP experiments were very illustrative and the correspondence between the licking pattern, the place field representation changes, and the conceptual state diagram is very striking. An interesting future experiment would be this:

—> Convert the reward to a stimulus instead. Eg. A Novel stimulus is superposed at either near indicator region, or far indicator region. There is no reward, except for knowing "where there surprising stimulus will occur". It'd be interesting to see whether this is sufficient to drive the decorrelation, instead of an actual reward. Maybe this could be mentioned in the supplement, or in the main paper if there is space.

Line 287 - 289: I agree. Often theoreticians and experimentalists worry that single-cell tunings cannot be interpreted, and that the information is "mixed selectivity" or "distributed coding". This is an experiment clearly showing that single cell tuning, properly interpreted in the context of a latent code, makes sense, and can even help understand the learning dynamics at a

fine granularity.

Lines 341-347

The set of observations here, and the corresponding follow up in the computational models is exemplary.

The careful comparison of CSCG trained using Expectation Maximization (EM) and the Hebbian RNN, is quite interesting and has generated a good number of questions that theoreticians and experimentalists can explore. There are indeed a fundamental differences between an RNN type representation (even with Hebbian learning rule) and an HMM-like representation trained using EM. RNNs do not decouple the latent dynamics from the observed. This can lead to better accuracy, but at the expense of posterior inference that is useful for planning and behavior. Also CSCGs bottleneck information through a discrete representation, which might also make it more amenable for planning.

It is quite interesting to me that this difference is visible in the experiments conducted on animals! This detail could have easily been overlooked, and the paper would still have been a strong one, but the fact that the authors paid attention to this and followed up is commendable. The significance of this should be pointed out in the discussion section.

The observations regarding transformers and LSTMs are also extremely interesting. Gradient-based end-to-end learning in expressive networks can learn myriad of ways to do the prediction task, but those representations are not good for driving the behavior of an agent using planning. This debate is starting to become central in AI, and maybe the authors could weave this into the discussion section as well. The comparison studies appeared well done to me, with an appropriate number of regularization techniques tried. The competing models did not look artificially handicapped.

Experiments on the novel tasks:

These experiments are very interesting too. Interestingly, the authors did not do the corresponding computational counterparts of these experiments. I do not think it is necessary, because the paper is already excellent and has performed a sufficient array of experiments to support the main points, but this is something that stands out as interesting future computational work. Coincidentally, the CSCG model authors have published a "fast rebinding" version of their model that addresses exactly one of the experimental scenarios described here. <https://openreview.net/pdf?id=3AreDQZ8eO> I think this is worth mentioning and citing. Also, this paper from Jadhav lab [https://www.cell.com/cell-reports/pdf/S2211-1247\(23\)00257-7.pdf](https://www.cell.com/cell-reports/pdf/S2211-1247(23)00257-7.pdf) looks relevant enough to be cited in context.

Version 2:

Reviewer comments:

Referee #1

(Remarks to the Author)

This is an admirable revision. The manuscript has seen many noticeable improvements. The new exposition on stretched environments, the better quantification of the learning-related features in the different models, but mainly for me, the improved exposition of the diversity of response patterns that are integrated in the orthogonalization process are helping this manuscript stand out further. This new figure allows the beautiful statement "that cognitive map formation is not a simple accumulation of (place and splitter cells) phenomena but a systematic, stereotypical progression in neural representations". I don't see any further issue with the manuscript.

(Remarks on code availability)

I would note that this code does not pretend (nor should it) to be a usable resource to the community.

I did not install and run the code, but read through it to figure out some particular implementation detail. I think this will be the main reason why researchers would like to have access to these scripts anyway. The code is well organized and the information is easy to extract. There is a README file, but it is rather short. The authors instead provide a notebook style organization. It is overall quite appropriate.

Referee #2

(Remarks to the Author)

I am satisfied with the responses to my comments.

(Remarks on code availability)

Referee #3

(Remarks to the Author)

I carefully read the new manuscript, and the response to all the reviewers, and also looked at the tools and code. I think the

authors have done an outstanding job addressing the concerns all the reviewers raised. The interactive visualization tool and tutorial are helpful for readers to explore in depth.

The additional modeling work has strengthened the manuscript further, and I like the further clarification in the manuscript that it is not a distinction between 'gradient based' vs not because EM is also gradient based (fixed point updates).

Overall I think this will be a highly impactful paper steering the field in the right direction. I support publication.

(Remarks on code availability)

I have only looked through the code, haven't actually run it. It appeared that code and Jupyter notebooks for reproducing the figures in the paper are provided in an easy to use manner.

We thank the referees for their encouraging and thoughtful assessment of our work. We found all the referees' critiques very constructive and have addressed them through extensive data analysis, new simulations, and careful revision to the text.

Revisions to the paper are substantial. Therefore, we did not highlight the changed text. To orient referees toward the largest changes:

- Figures 3 and 4 have been significantly expanded, with new analyses of single-cell tuning properties and additional modeling results and quantifications.
- All main text sections have been substantially revised to address referees' comments and incorporate new insights.
- Multiple Extended Data figures are updated to address various aspects of the referees' comments.
- We have introduced an interactive visualization tool (<https://cognitivemap.janelia.org/>) accompanied by a narrated tutorial (<https://youtu.be/kxg3eAG1uXk>) for using this tool, to facilitate the readers' exploration of the data, enhancing the accessibility of our findings.

These changes provide a more nuanced understanding of hippocampal cognitive map formation process and strengthen the paper's contributions to the field. We believe these revisions address the referees' concerns and significantly enhance the overall quality and impact of our work.

We begin our response by addressing three major points that were brought up by multiple referees or that seemed of general interest.

1. Further clarification of the computational principles underlying orthogonalized representations and learning dynamics.
2. Additional quantification for the co-evolution of behavior and neural activity as well as the comparisons to the modeling results.
3. Better articulation of the novel contributions that extend beyond splitter cells and pattern separation.

After responding to these major points, we include a point-by-point response to the individual referees.

Point 1: Computational principles underlying orthogonalized representations and learning dynamics

In our original manuscript, we showed that both CSCG (a type of HMM) and an RNN with Hebbian plasticity can learn the orthogonal representations of the two trial types, as observed in the mouse hippocampus. In contrast, LSTMs and Transformers do not naturally orthogonalize; instead, they represent the two trial types using small differences in hidden unit activations that are amplified by the output weights for accurate predictions. In other words, these models all achieve good performance in next-input prediction, but they do so using very different representational structures (*i.e.*, orthogonal vs correlated). The referees appropriately encouraged us to more deeply explore the causes behind these differences. To do so, we performed additional numerical simulations and analyses. Our findings revealed that the emergence of decorrelated representations is influenced by a combination of factors, including the model's architectural design, learning objectives, and algorithms employed. Specifically, we trained vanilla RNNs using backpropagation through time (BPTT) and discovered they can produce orthogonalized representations of the task when using "soft Winner-Take-All" (sWTA) activation functions, such as normalized exponential or high-power polynomial functions (both can be considered as softmax activation functions), which enhances existing difference in unit activations. In contrast, ReLU or sigmoid activation functions do not produce orthogonal representations (New Figure 4 and Extended Data Fig. 13).

These new results, combined with our original findings, suggest that orthogonal representations can emerge through multiple mechanisms: (1) learning objective: direct regularization of the output's correlation structure (e.g., adding correlation penalties to the cost function of LSTMs, Extended Data Fig. 13), (2) specific architectural features like the choice of softmax activation functions, and (3) learning algorithms that inherently promote orthogonalization, such as Expectation-Maximization (EM) in CSCGs.

The CSCG model is particularly interesting because it naturally promotes orthogonalization through the EM algorithm. By assigning different "clones" (hidden states) to represent the same sensory input in different contexts, CSCGs can maximize the likelihood of observed sequences. This allows the model to better predict future sensory inputs based on context, even when current inputs are ambiguous. As learning progresses, the EM algorithm gradually shifts probability mass to context-specific clones, resulting in orthogonalized representations that efficiently capture the task structure. To illustrate this process more concretely, we have provided a simple example case that demonstrates why CSCGs naturally lead to orthogonalized representations (Extended Data Fig. 11).

While the final orthogonalized representations are important, our study emphasizes that each of the steps in the sequence of learning to reach these representations is equally mechanistically informative. Our ability to image neural activity longitudinally, over several days as mice learned

the task, allowed us to determine the exact sequence of orthogonalization steps during learning. We observed that correlations of the off-diagonal neutral zones disappeared first, followed by decorrelation of the neutral zone preceding the far reward cue (Pre-R2), and finally decorrelation of the neutral zone preceding the near reward cue (Pre-R1). This sequence provided an important constraint for evaluating computational models. Interestingly, while several models captured the final orthogonalized representations, only the CSCG model accurately replicated both the final state and the specific learning trajectory observed in animals. Other models, including Hebbian-RNN and vanilla RNNs with softmax activation functions, failed to reproduce the correct learning sequence despite achieving the final orthogonalized state (new Figure 4, Extended Data Fig. 13).

Our findings indicate that the learned orthogonal representations and the exact decorrelation sequence to reach these representations are not ubiquitous among the models we studied. Specific choices about model architecture and algorithmic implementation determine whether models exhibit representational structure and learning dynamics similar to those seen in the animals. Based on our current evidence, CSCG stands out as the best-matching model, suggesting that cloned Hidden Markov Models trained using the EM algorithm could be a promising candidate for understanding structure learning in the hippocampus. The unique ability of CSCGs to capture both the final orthogonalized state and the specific learning trajectory highlights their potential as a powerful tool for modeling hippocampal function. At the same time, we recognize that hippocampal learning is complex and multifaceted. Future research may reveal additional critical mechanisms, such as the role of reward signals in shaping cognitive map formation. We are grateful that the referees pushed us to better understand these complex phenomena.

Point 2: Additional quantifications for experimental data and modeling results

The referees highlighted the need for more quantitative analyses in our study, particularly regarding the quantitative comparisons of the representations learned by the animals and the models as well as the correlation between behavioral states and neural decorrelation. To address these concerns, we have now provided two new analyses:

1. Quantitative comparisons of representations between animals and models: In the updated Figure 4, we have included numerous new models and the quantifications for the late-stage correlation and the sequence of orthogonalization for key track regions for different models. We show that the average late-stage correlation is low in the animal and more aligned with the final representations of CSCG, Hebbian-RNN, vanilla RNN with exponential or polynomial softmax activation functions. In contrast, late-stage representations in vanilla RNN with ReLU or Sigmoid activation, LSTM, and Transformer do not conform to those of the animals (New Figure 4f, Extended Data Fig. 13). Additionally, only CSCG exhibits a similar sequence of decorrelation to the animal (Pre-R2 followed by Pre-R1, New Figure 4d, j). These quantifications enable objective comparisons of the learned representations between the animal and different models.

2. The correlation between behavioral changes and neural activity decorrelation dynamics:
We performed correlation analysis between the CPD scores for multiple behavior strategies and the patterns of neural activity correlation (key regions in the neural correlation matrix). We observed significant correlation between changes in the behavioral coefficients and the decorrelation of neural activity through learning (Extended Data Fig. 8).

We believe these additional quantifications address the referees' concerns by offering precise comparisons between animal and model representations. Moreover, they reinforce our argument that hippocampal neural representations co-evolve with changes in behavioral strategies.

Point 3: Novel contributions beyond splitter cells and pattern separation

We appreciate the referees' comments regarding our study's relationship to previous work on splitter cells and pattern separation. While these concepts form an important foundation, our work makes several novel contributions that significantly advance our understanding of hippocampal function and cognitive map formation.

First, our study provides a comprehensive, longitudinal view of hippocampal neural dynamics during learning by imaging thousands of neurons over weeks. We show that cognitive map formation is not a simple accumulation of splitter cells or gradual increase in pattern separation. Rather, the depth of our dataset reveals a systematic, step-by-step progression in neural representations that underlies cognitive map formation. Our analysis demonstrates that splitter cells emerge selectively at specific track locations corresponding to task-relevant decision points or ambiguous states. This targeted emergence, as opposed to uniform accumulation, reflects the hippocampus's efficient representation of task structure.

Second, our analysis extends beyond traditional cell-type classifications, revealing a diverse spectrum of single-neuron tuning properties. In our new analysis (new Figure 3), we now characterize the prevalence of multiple single cell *response types*, including regular splitters, remapping splitters, place-like responses, as well as mixed phenotypes. We quantify these using a two-dimensional feature space that plots the correlation of tuning between *Near* and *Far* trials against a difference score measuring the disparity in peak activity. This approach reveals a continuum of response properties rather than discrete categories. Our data show how individual neurons dynamically transition between these functional roles as learning progresses. The changing prevalence of these response types across different track locations and learning stages provides insight into how the hippocampus adapts its representations at the single-neuron level to capture task-relevant information. For example, we observe that cells tuned to the reward and pre-reward regions gradually transition from place-like responses to splitter responses, indicating that these sensory-ambiguous regions require more prolonged learning to produce differential firing patterns. To facilitate exploration of this rich dataset, we have provided an interactive data

visualization tool that allows readers to browse individual cell tuning data across learning stages and track positions, along with a narrated tutorial for using this tool.

Third, our study bridges the gap between cellular-level observations and computational theories of hippocampal function. By comparing our data to various computational models, we show that the observed neural dynamics are best explained by latent state inference processes. This finding provides strong empirical support for theoretical proposals about the hippocampus's role in discovering hidden task structure, moving beyond simple tuning associations or pattern separation.

Fourth, our work reveals a previously unrecognized property of hippocampal learning: the specific sequence of decorrelation in neural representations. We show that different parts of the environment become decorrelated in a particular order, a finding that provides crucial constraints on computational models of hippocampal function. Importantly, our analysis demonstrates that only the CSCG model, which performs explicit state inference, captures this specific learning trajectory. This unique capability of CSCGs offers new insights into the algorithms the brain might use to form cognitive maps. Our findings suggest that the formation of the orthogonalized state machine representation we observe is not merely a consequence of pattern separation, but a highly structured process producing a code that captures the latent task structure.

In summary, while our work builds on concepts like splitter cells and pattern separation, it also provides a fundamentally new understanding of hippocampal function.

Point-by-point responses to the referees (Referees' comments in bold Arial font, our responses in regular Times New Roman font):

Referee #1 (Remarks to the Author):

The manuscript “Learning produces a hippocampal cognitive map in the form of an orthogonalized state machine” reports a longitudinal observation of thousands of hippocampal cells as animals learn to operate in a two virtual environments. The number of cells observed, while still only about 1-2% of the total number of neurons in the area, is staggeringly high, especially for a longitudinal study. By diving into this large data, the authors found that the hippocampus gradually represents distinct states with different response patterns. This decorrelation appears to reflect the learning behavior. The authors then asked whether such a feature is unusual or not, by training different machine learning and neuroscience models and testing whether the hidden states follow this tendency to decorrelate representation through learning. They found that transformers and LSTMs trained with backprop did not while a particular type of hidden Markov model and a Hebbian spiking neural network did.

The study addresses a crucial question of interest to most neuroscientists: how does

learning proceeds in a large population? I think this paper contribute two rather important observations. One is that the learning of the cognitive maps is associated with a stage-like decorrelation of the representation, as both the behavior and the representations appear to switch between relatively stable states. The other is that the gradual process of decorrelation is uncommon in sequence learning algorithms, but natural in a cognitive models and spiking neural networks. These findings are consistent with a lot of what we know of the hippocampus and machine learning models. I don't know whether the findings will be considered as obvious or not to most readers, but I think it is steering the field in the right direction.

The manuscript is particularly well written. There is an attention to detail and a clarity of the exposition that is commendable. The unusually long discussion makes the findings more salient by making well-read comparisons with various parts of the literature. The data appears to me of excellent quality. The analysis and modeling would, however, require further attention (see below).

We thank the referee for these encouraging comments.

Some aspects require further attention:

1. What is being compared? The article is framed around the question of learning. Comparing the features found in the data with computational models is could shed light on the brain's learning algorithm. The failure of LSTMs and transformers is flagged with the 'gradient based' aspect of their learning algorithms. However, across the models considered in the article, many conceptually relevant aspects change: architecture, model goal (e.g. a model of cognition or a model for sequence learning) and learning algorithms (e.g. STDP, backprop, EM). I find the paper is vague about what is the main conclusion of this comparison. Is it simply to show that decorrelation is not trivial? Or is it to conclude that we should take one algorithm over another for the hippocampus and why?

We thank the referee for this important feedback. We acknowledge that our original framing of the comparison between models as "gradient-based vs. non-gradient-based" was oversimplified and potentially misleading. As the referee correctly points out, many aspects vary across the models we considered, including architecture, learning goals, and learning algorithms.

To address this, we have conducted new simulations and analyses, as detailed in Point 1 of our major responses. Our findings reveal that the emergence of orthogonalized representations is influenced by multiple factors, including the specific architectural design of the model (such as activation functions), learning rules, and learning objectives. The interplay of these elements, rather than any single factor, appears to drive the development of representations that resemble those observed in the hippocampus.

Importantly, we found that while several models can achieve orthogonalized representations, including some gradient-based models with appropriate activation functions, CSCG remains the only model that matches both the final orthogonalized representations and the specific learning sequence observed in animals.

We have modified the main text to highlight these findings, emphasizing that hidden state inference from ambiguous sensory sequences, as exemplified by CSCG trained using the Baum-Welch EM algorithm, could be a fundamental computational principle underlying hippocampal structure learning. Additionally, we have included discussion about potential alternative mechanisms (e.g., reward-driven learning) to provide a balanced perspective on our findings.

We appreciate the referee's critique, which has led to a more nuanced and accurate presentation of our modeling results.

1.2 LSTMs and Transformers are tagged as ‘gradient based’ in opposition to the other two models, implying that this aspect of the learning algorithm is what distinguishes these models. However, EM is also a gradient-based algorithm as it is a method to maximize the likelihood by an approximation that switches between two categories of parameters, iterating between two gradient descent steps. EM may not be the backpropagation of error algorithm, but maybe backprop is not the right algorithm for cloned HMMs. In any case, the emphasis on gradient-based as a distinguishing factor seems ill-placed.

We thank the referee for this insightful feedback. We agree that our original emphasis on "gradient-based" as a distinguishing factor was misplaced, and we appreciate the opportunity to clarify this point.

Our additional analyses reveal that the ability to achieve orthogonalized representations is not simply determined by whether the learning algorithm is gradient-based or not. Instead, we found that orthogonalization emerges from an interplay of factors, including architectural features, learning objectives, and learning algorithms employed. For instance, we found that RNNs trained with gradient-based BPTT can achieve orthogonalized representations when using softmax activation functions, but not with ReLU or sigmoid activations.

The EM algorithm used in CSCGs, while different from backpropagation, is indeed related to gradient-based methods as the referee points out. Both can, in principle, reach similar solutions, although their learning trajectories appear to differ.

These findings suggest that certain network architectures naturally promote decorrelated representations, regardless of the specific learning rule employed (e.g., Hebbian RNNs vs Vanilla RNNs trained via BPTT). The key factor appears to be the ability to magnify small differences in

input, which can be achieved through activation functions such as softmax or the EM algorithm (New Figure 4 and Extended Data Fig. 11).

We have revised the manuscript to reflect this more nuanced understanding, removing the emphasis on "gradient-based" as a distinguishing factor and instead focusing on the multiple factors, including architectural features, learning objectives and algorithms, in producing the orthogonalized representations. We thank the referee for this important feedback.

1.3 In the context of what is being compared, the STDP RNN choice is also confusing. One would assume that the authors wanted to include a bottom-up model of the hippocampus. But the authors will agree that a RNN is the wrong model for CA1. Perhaps the authors wanted to include the RNN as a model of CA3, but it should really be made explicit (by adding a readout layer and calculating the PV-correlation on the readout layer). Note that Diehl and Cook studied a similar model as Kappel and Maass but without recurrence, which would be appropriate here.

We thank the referee for their insightful comment about the Hebbian-RNN model. This has prompted us to clarify our modeling approach and its relationship to hippocampal circuitry. The Hebbian-RNN in our study is indeed intended to model CA3 rather than CA1. We have now made this explicit in the paper to avoid any confusion. We made an implicit assumption that CA1 could inherit orthogonalized responses from upstream regions. To support this assumption, we have conducted additional simulations with a readout layer (with varying levels of sparsity) following the recurrent layer in Hebbian-RNN. These simulations demonstrate that the readout layer inherits orthogonal responses from the recurrent layer (Extended Data Fig.13 d, e).

Our primary goal with these models is to explore which architectures and learning principles can produce orthogonalized representations like those we observed experimentally, rather than to create a detailed model of hippocampal circuitry. We agree that decorrelation via STDP and "soft winner-take-all" dynamics doesn't necessarily require recurrence, as demonstrated by Diehl and Cook. However, our task involves sequential prediction, which may benefit from recurrent connections that can encode transition probabilities. We agree with the referee that exploring the learning representations and dynamics in pure feedforward networks is an important future direction. We have revised the manuscript to make these points clearer, explicitly stating the intended correspondence between model components and hippocampal subregions. We thank the referee for prompting this clarification.

1.4 The comparison sits on the PV-correlation across near and far, both in algorithms and data. While the data samples from CA1 only (and not from CA3, DG or dorsal visual cortex), where are the models sampled from? I could not find any indication of this in the methods. It would seem that sampling from the readout layer or the hidden states of a transformer would yield widely different results. Transformers and LSTMs not being models of the

hippocampus, there is not logical place to constrain the comparison, yet this can have implications on the resulting correlation structure.

We have addressed the lack of clarity in our methods section regarding the layers used for representation comparison by adding the necessary details in the revised manuscript. In our analysis, we aimed to examine layers in each model that we believed would best capture the learned task representations. For LSTMs, we analyzed the activity of the hidden units, while for Transformers, we examined the pre-logit layer as it represents the final stage of feature extraction before classification, potentially capturing the most task-relevant information. For Hebbian/Vanilla RNNs and CSCG, we studied the activity of the hidden units and clones. However, we recognize that the choice of which layer to analyze, particularly in complex models like LSTMs and Transformers, is not straightforward and could impact the results. To further explore this, we have now also analyzed the cell states in the LSTM. Our results show that cell states also do not exhibit the orthogonal representational structure observed in the hippocampal data (Extended Data Fig. 13a). We have now revised the methods section to explicitly state which layers were analyzed for each model.

2. Stage-like? There are many mentions of stage-like learning, both in terms of representation and behavior. However, I could not find any quantification of this. This is not a central claim of the paper (although it is an interesting observation). The paper nonetheless contains claims that should be supported by evidence or removed. Behaviour seems stage like, but it is not quantified. Representations gradually decorrelate but stage-like would imply a significant difference between the progression of Pre-R1 and Pre-R2 in Fig 2f, which seems to be the case but is not tested. Ultimately, whether the stages in representations correspond to stages in behavior is not tested either.

We appreciate the referee's observation about our use of "stage-like" terminology. We acknowledge that our original presentation may have overstated the discreteness of these changes. To address this, we've revised our language throughout the manuscript to emphasize the gradual nature of the observed changes in both behavior and neural representations. For example, in the result section we have added:

“These strategies represent predominant behaviors that emerge and fade gradually, rather than discrete, abrupt changes.”

To quantify the relationship between behavioral changes and neural activity, we've conducted new correlation analyses. We examined the relationship between neural activity and behavior by performing correlation analysis on the Coefficient of Partial Determination (CPD) scores for behavioral strategies and the patterns of neural activity. These analyses reveal significant correlations between changes in behavioral metrics and the extent of decorrelation in neural activity (new Extended Data Fig. 8).

3. Transformers. While LSTMs is a logical choice for sequence learning, the choice of transformers is less clear. I also found that the description in the methods was insufficient to understand how the transformers were implemented. In particular, I am puzzled by how the authors forced the algorithm to receive one input location at a time. Transformers normally spatialize sequences and need to wait for the whole sequence to provide an output, a feature that would appear to be antagonistic with the requirement of the task.

We thank the referee for highlighting the lack of clarity in our transformer implementation. We have revised the methods section to include important simulation details. To address input handling, we trained transformers with context lengths from 1 to 100, finding that lengths exceeding 4 solved the task. This threshold is task-specific, enabling disambiguation among reward locations given the inter-reward gray cue length of 3. While transformer training differs from other models by using sequence segments as inputs, we included these results due to transformers' relevance in sequence learning and prediction tasks. Our findings suggest that even with full task context, self-attention operations do not lead to orthogonalized representations. We've clarified these points in the revised manuscript to improve understanding of our transformer implementation and its implications.

4. Splitter cells. Is the decorrelation observed 100% synonymous with the concept of splitter cells? If you'd remove all the splitter cells, what correlation patterns would remain? This is an important question to assess the novelty of the experimental findings.

We appreciate the referee's question about the relationship between decorrelation and splitter cells. Our new analysis, presented in Figure 3, quantifies the prevalence of different response types. Note that we avoid describing them as “cell types”, because response types are plastic, and therefore not necessarily a stable feature of any given cell. We now use a 2D feature map plotting near vs far trial tuning correlation against difference scores (new Figure 3d), revealing a continuous spectrum of cell response properties rather than discrete categories. This analysis shows how cells gradually transition from place-like responses to various types of splitter responses as learning progresses, particularly at positions where information allows for latent state discovery. The decorrelation we observe indeed aligns with the production of splitter cells. In the expert stage, for the track regions not including the beginning or the end, most significantly active cells are splitter cells, removing them eliminated nearly all active neural activity and results in low correlation due to uncorrelated noise in silent cells. In earlier sessions, removing cells with splitter responses increases correlation, as the remaining cells are not yet trial-selective. Regarding novelty, our study goes beyond simply identifying splitter responses. We reveal a diverse spectrum of neural tuning properties and show how cells dynamically transition between functional roles during learning. This comprehensive, longitudinal view provides new insights into the process of cognitive map formation, as detailed in Point 3 of our major responses.

5. Quantification. There seem to be no quantification of the difference between mouse and the models. The heat maps are presented as qualitative evidence.

We thank the referee for highlighting the need for quantitative comparisons between our experimental data and model results. In response, we have performed multiple quantitative analysis in the new Figure 4, as we have outlined in major response point 2. We quantify both the degree of orthogonalization in the final learned state and the sequence of decorrelation during learning for key track regions. These quantifications reveal that while several models can achieve orthogonalized representations like those observed in mice, only the CSCG model accurately captures both the final state and the specific sequence of decorrelation. This more rigorous comparison provides a stronger basis for our conclusions about which computational principles best explain the observed hippocampal dynamics.

Minor

6. The supplementary figure on STDP+RNN is not consistent as different orderings of activity shows dramatically different levels of activity. (E.g. the near sorting in the near trial shows a diagonal, but the near trial in the far sorting shows mostly white; we would expect scattered activity, unless there is just very little activity on that time scale). It is not clear to me that this network requires the recurrent excitatory connections to capture the decorrelation. Diehl and Cook showed surprising results based on Kappell et al. but in the absence of recurrent connection (and not for HMMs).

In our original Supplementary Figure 11b, we only displayed the most active cells for *Near* and *Far* trials separately. This approach was chosen because, as learning progresses, cells increasingly show differential firing between the two trial types. Consequently, when sorting cells based on activity in one trial type and displaying their activity in the other, many cells appear inactive at the expert stage. We now recognize that this presentation method was inconsistent with our main Figure 2 c and d, which shows all cells' activity regardless of their activation levels. To address this discrepancy and provide a more comprehensive and accurate representation, we have revised Supplementary Figure 11 (now Extended Data Figure 12) to include an additional activity plot including all cells for both trial types. This revision allows for a direct comparison with the main figures. Regarding the necessity of recurrent connections, we acknowledge that Diehl and Cook (2015) demonstrated successful MNIST classification using STDP in a feedforward network. However, their task involved learning to classify static digit images, with each image presented independently. In contrast, our task differs fundamentally as it requires learning both sensory inputs and transition dynamics to build a latent graph. This complexity necessitates context retention across time steps, which is naturally achieved through recurrent connections. While some previous studies have explored feedforward networks for learning state transitions using long-lasting sensory input traces, their ability to produce the specific decorrelation patterns we observe

remains uncertain. Recurrent connections provide a mechanistic basis for maintaining and updating context, crucial for disambiguating identical sensory inputs in different task states.

We acknowledge that alternative architectures may achieve similar results. Exploring the potential for feedforward networks to provide necessary context signals without recurrent weights is an intriguing avenue for future research. We thank the referee for prompting this clarification.

7. Germane to the idea of spotting learning rules by looking at the evolution in neuronal activity are: Nayebi et al. Neurips “Identifying learning rules from neural network observables” (2020), Wang et al., BioRxiv “inferring plasticity rules from single-neurons” (2023); the present paper can be seen as a feature-matching approach instead of ML classification.

We appreciate the referee highlighting these relevant papers. Our approach aligns with a 'feature-matching' method for inferring learning rules from neural activity dynamics. We have added these references to our discussion, noting: "Our approach can be viewed as a feature-matching method for inferring learning rules from neural activity dynamics, complementing recent work on identifying learning rules from neural observables [Nayebi et al., 2020; Wang et al., 2023]."

8. Other concepts in machine learning may relate to this question. I am thinking of contrastive losses (also related to neuroscience now, see Halgaval and Zenke Nature Neurosci (2023), which are more naturally including the concept of orthogonalization. Another concept in machine learning is object- and agent-centric representations. These ideas may lie beyond the scope of the present work, but it is likely that these techniques capture decorrelation in a natural way while using backprop for training.

We thank the referee for highlighting these important concepts and relevant literature. We agree that contrastive losses and object- and agent-centric representations are highly relevant to our work and that these techniques can indeed capture decorrelation naturally while using backpropagation for training.

In response to this valuable suggestion, we have added the following discussion to our manuscript:

“In addition to these biological mechanisms, recent machine learning advances offer new perspectives on achieving decorrelation through backpropagation-like processes. Techniques such as contrastive losses and object-centric representations provide alternative approaches to generating decorrelated representations, which may have parallels in biological learning systems.”

9. A small thing to check, but it seems that the LSTMs would have a different match if the labels for a given position were not repeated multiple time ([1,2...]) instead of

[1,1,1,1,2,2,2,2...].

We thank the referee for this insightful suggestion. In our study, we intentionally chose an aliased task by using repeated sensory inputs to represent different latent states, mirroring the ambiguity animals face in the actual task environment.

We conducted additional experiments training LSTMs on sequences where each position had a unique sensory representation ([1,2,3...] instead of [1,1,1,1,2,2,2,2...]). Our observations showed that this indeed led to increased decorrelation of states within each trial type. However, high diagonal correlations persisted between the two trial types because the sensory inputs at corresponding positions were still shared across trial types, again suggesting LSTM do not naturally orthogonalize aliased sensory inputs corresponding to distinct task states.

Referee #2 (Remarks to the Author):

In this paper, Sun and colleagues propose that area CA1 of the hippocampus supports learning of an "orthogonalized state machine", where representations of task states become progressively decorrelated. They present evidence for this proposal using calcium imaging data of many CA1 neurons in mice. Their modeling shows that only an HMM-like model acquires the decorrelated representations characteristic of the neural data.

Overall, I thought this was an interesting and well-written paper, though I'm not sure it represents a huge leap beyond previous data. The idea that the hippocampus does some form of structure learning, and is capable of pattern separation, is well-established (as the authors note). What's new here is the large-scale and longitudinal aspect of the data, which allowed the authors to reveal the gradual emergence of structure. I found that aspect novel and exciting.

We thank the referee for these comments.

I felt that the paper was limited in a few ways. First, the experimental design was rather unimaginative relative to the complexity of the models under consideration.

We appreciate the feedback regarding the experimental design. Our primary goal was to strike a balance between creating a task complex enough that requires substantial structural learning (e.g., aliased sensory segments within and across trial types) and a task simple enough for the mice to learn within a few weeks. We also aimed for the mice to discover the task on their own with minimal supervision, avoiding punishment and limiting behavior shaping. We believe this approach achieved our objectives, enabling us to conduct experiments that yielded valuable insights into cognitive map formation. The relative simplicity of the task allowed us to focus on the underlying learning process and neural mechanisms, which are central to our study. This design was also practical, considering the technical challenges associated with long-term imaging in mice

and the need to study a sufficient number of animals for robust findings. By carefully selecting task complexity, we captured the full learning trajectory within a timeframe suitable for longitudinal studies while maximizing our sample size.

Second, the approach to modeling was not as comprehensive as one might like; there was no evaluation (as far as I could tell of) parameter sensitivity, no model fitting, no formal model comparison. The interpretation rests almost entirely on qualitative visual inspection.

We thank the referee for pointing out the need for a more comprehensive modeling approach. We have now addressed this by adding several new quantitative comparisons of the data and the models, as described in our major response point 2 and response to referee 1. Specifically, we have included new quantifications in Figure 4 that compare the final orthogonalized representations and the learning sequence across different models. These additions provide a quantitative basis for our interpretations.

Our modeling results are robust across a wide range of parameters, including learning rates, number of units/clones, and choice of optimizers. However, certain factors, such as the choice of activation functions for RNNs and the positioning of reward symbols, significantly influence learning dynamics and representations (new Figure 4, Extended Data Fig. 10 and 13). Rather than optimizing for the best possible fit, we focused on understanding the computational principles that can capture key features of hippocampal learning, such as the specific sequence of decorrelation and the final orthogonalized state.

Third, and more importantly, there was no attempt to elucidate the underlying principles explaining why some models match better than others. The considered models are very different from each other; to explore the necessary and sufficient conditions for matching neural data, the authors would need to develop systematic variants of the winning model.

We appreciate this important feedback, which aligns with feedback from other referees. We have conducted substantial additional modeling to address this important point, as detailed in our major response 1 and our reply to referee 1. Our expanded analysis reveals that the emergence of orthogonalized representations matching our neural data is influenced by an interplay of factors, including the model's architectural design, learning objectives, algorithms employed, and input sequence structure. We explored this by examining various RNN architectures with different activation functions and investigating how CSCG results are affected by specific sequence designs, such as the order of reward and sensory cues (Extended Data Figure 10). These results provide additional insights into the conditions that lead to representations matching our neural data. While several models can achieve orthogonalized final representations, our current results suggest that the CSCG model best captures both the final state and the specific learning trajectory observed in animals. This indicates that models capable of hidden state inference from ambiguous sensory sequences may be particularly relevant for understanding hippocampal learning dynamics.

Finally, the authors do not evaluate any of the models on the test conditions (new cues and stretched trials). I suspect this evaluation would be highly informative about the adequacy of the models.

We sincerely appreciate the referee's suggestion to evaluate our models on the test conditions. This is indeed a valuable direction that could provide further insights into the adequacy of our models. We have begun preliminary explorations with CSCG and RNNs, and our initial results reveal complex patterns that warrant deeper investigation. However, fully modeling these scenarios would require substantial expansions to our computational frameworks, extending beyond our current focus on initial cognitive map formation. Given the complexity and potential significance of these simulations, we believe they would be best addressed in a dedicated follow-up study. We look forward to exploring this avenue in future research.

Minor comments:

In the figure captions, please state what the error bars show.

Fixed.

p. 12: "representational structure of cognitive map" -> "representational structure of the cognitive map"

Fixed.

p. 30: "used Viterbi training algorithm" -> "used the Viterbi training algorithm"

Fixed.

Referee #3 (Remarks to the Author):

I thoroughly enjoyed reading this paper. The findings are important not only for the neuroscience community, but also for the community of AI researchers that are looking to the neuroscience field for inspiration. The paper has major strengths in several aspects, and is a well done study that I'd strongly recommend for publication in Nature.

We thank the referee for these comments on our work and its potential impact.

This is one of the first experiments that brings out the role of perceptual ambiguity, and how the different stages of learning in the hippocampus overcome this perceptual ambiguity in an experimental setting in a proper way. Animals get only sensations from the world — they do not get GPS coordinates. How an animal recovers representations like “locations” from this egocentric sensory experience is one of the main mysteries behind hippocampus representations. The experimental setting in this paper goes at the heart of this question, more thoroughly than previous studies. Earlier studies were limited by the number of neurons recorded or by the ability to track them across long durations. This paper overcame

those limitations to perform a set of well designed experiments that ask and answer important questions in the field while also opening up new and interesting directions for future investigations.

Moreover, their computational studies are well done and powerful. One drawback of many experimental neuroscience papers is that they experimental neuroscientists often hand-craft a model based on a theory they arrived from the paper.

Refreshingly, this paper is doing the opposite. They have the observations, but they don't try to fit a new theory, or add bells and whistles to an existing model. Instead, they rigorously test how the model compares against well known paradigms in machine learning, one of which also include a model called CSCG, an ML model that is also gaining attention as model of the hippocampus. The studies are well done, with proper controls, regularizations, and observations. and the conclusions are important for multiple communities. One of the striking aspects of the paper to me was that the researchers paid careful attention to their experiments to pick up learning dynamics differences that are in some ways central to some of the debates on internal representations, world models, uncertainty, and planning. I mention some of the details of this in the detailed review below.

We appreciate and thank the referee for these kind comments putting our work in perspective of the bigger field.

One model that is missing from the comparisons is the Tolman-Eichenbaum Machine (TEM), and readers might be curious why. It would be good to address this in the manuscript itself. I do not think a new experiment is necessary. It might be worth mentioning that TEM doesn't deal with perceptual ambiguity from exposure to one environment. It needs exposure to multiple environments sharing the same structure to overcome the aliasing of the kind in this experimental setting. Moreover, TEM also requires an allocentric action map, which is not the case with the teleportation setting in this experiment — this could be another reason why TEM might not be applicable. Similarly, Successor Representations (SR) is another model that is popular among hippocampus researchers. I think it is worth pointing out to the readers that SR doesn't deal with perceptual ambiguity and is only applicable in settings where locations are assumed to be directly sensed. Again, no new experiment is necessary because the SR model is simply not applicable in this setting.

We thank the referee for this suggestion. We've added a discussion in the manuscript addressing why we didn't include TEM or SR in our comparisons:

"While models like Successor Representations (SR) and the Tolman-Eichenbaum Machine (TEM) are valuable frameworks in modeling cognitive maps, we focused on models that directly capture learning dynamics in our single-environment task. TEM primarily addresses cross-environment generalization and SR assumes directly sensed locations, which differ from our main single-task setting and its perceptual ambiguity."

This concise explanation addresses the key points about TEM and SR's limitations for our specific study while acknowledging their broader importance in hippocampal modeling.

More detailed comments from the read through.

Line 107: “short term memory of the indicator cue after it disappears “

Yes, this is an important point. But it might also be worth point out at this stage that, as it turns out, this short term memory is carried forward by creating latent states.

We thank referee this suggestion and added the following sentence in the results section:

“Within this learned structure, the short-term memory of indicator cues is carried forward by distinct neural activity representing different latent states.”

Figures 2c and 2d nicely indicate the sensory ambiguity that is reflected in the early learning stage, and then how they decorrelate after learning. It is impressive to see how the behavior, the hypothesized state diagram for the behavior, and the neural states correspond well at different stages of learning. It is also striking to see that it is possible to do this kinds of experiment that can really help us select between different theories. Similarly, the UMAP experiments were very illustrative and the correspondence between the licking pattern, the place field representation changes, and the conceptual state diagram is very striking. An interesting future experiment would be this:

—> Convert the reward to a stimulus instead. Eg. A Novel stimulus is superposed at either near indicator region, or far indicator region. There is no reward, except for knowing “where there surprising stimulus will occur”. It’d be interesting to see whether this Is sufficient to drive the decorrelation, instead of an actual reward. Maybe this could be mentioned in the supplement, or in the main paper if there is space.

We appreciate the referee's insightful suggestion to convert rewards to surprising stimuli. This approach could indeed help disentangle the roles of reward prediction and sensory prediction in forming orthogonalized latent states. We've incorporated this idea into our discussion:

"Future experiments replacing rewards with novel sensory cues could clarify whether reward is necessary for extracting latent task structure or if sensory prediction alone suffices. Such studies would distinguish between reward-driven learning and purely sensory-based predictive coding in cognitive map formation."

Line 287 - 289: I agree. Often theoreticians and experimentalists worry that single-cell tunings cannot be interpreted, and that the information is “mixed selectivity” or “distributed coding”. This is an experiment clearly showing that single cell tuning, properly interpreted in the context of a latent code, makes sense, and can even help understand the

learning dynamics at a fine granularity.

Lines 341-347

The set of observations here, and the corresponding follow up in the computational models is exemplary.

The careful comparison of CSCG trained using Expectation Maximization (EM) and the Hebbian RNN, is quite interesting and has generated a good number of questions that theoreticians and experimentalists can explore. There are indeed a fundamental differences between an RNN type representation (even with Hebbian learning rule) and an HMM-like representation trained using EM. RNNs do not decouple the latent dynamics from the observed. This can lead to better accuracy, but at the expense of posterior inference that is useful for planning and behavior. Also CSCGs bottleneck information through a discrete representation, which might also make it more amenable for planning.

It is quite interesting to me that this difference is visible in the experiments conducted on animals! This detail could have easily been overlooked, and the paper would still have been a strong one, but the fact that the authors paid attention to this and followed up is commendable. The significance of this should be pointed out in the discussion section.

We agree that the distinctions between CSCG and various RNN models, especially the distinct learning dynamics, deserve more attention. In response, we elaborated on how the CSCG model, with its HMM-like representation and discrete bottleneck, better captures the specific order of decorrelation observed in our animal data (Pre-R2 before Pre-R1). We have modified Figure 4 to highlight the match between animals and CSCG, adding annotations on new Figure 4d to highlight the specific sequence of learning. We also modified discussions to incorporate implications of this finding for our understanding of hippocampal computation.

We thank the referee for encouraging us to highlight this aspect of our results. We believe this addition will strengthen the impact and interpretability of our findings for both neuroscience and AI communities.

The observations regarding transformers and LSTMs are also extremely interesting. Gradient-based end-to-end learning in expressive networks can learn myriad of ways to do the prediction task, but those representations are not good for driving the behavior of an agent using planning. This debate is starting to become central in AI, and maybe the authors could weave this into the discussion section as well. he comparison studies appeared well done to me, with an appropriate number of regularization techniques tried. The competing models did not look artificially handicapped.

Experiments on the novel tasks:

These experiments are very interesting too. Interestingly, the authors did not do the corresponding computational counterparts of these experiments. I do not think it is necessary, because the paper is already excellent and has performed a sufficient array of experiments to support the main points, but this is something that stands out as interesting future computational work. Coincidentally, the CSCG model authors have published a “fast rebinding” version of their model that addresses exactly one of the experimental scenarios described here. <https://openreview.net/pdf?id=3AreDQZ8eO>; I think this is worth mentioning and citing. Also, this paper from Jadhav lab [https://www.cell.com/cell-reports/pdf/S2211-1247\(23\)00257-7.pdf](https://www.cell.com/cell-reports/pdf/S2211-1247(23)00257-7.pdf) looks relevant enough to be cited in context.

We thank the referee for pointing us to these very relevant references. We are indeed very excited to further explore CSCG and other model’s behavior in novel situations as follow up studies. We have now included these two papers in our references. We sincerely appreciate the referee’s thorough and insightful review.